# WHEN FLATNESS DOES (NOT) GUARANTEE ADVERSARIAL ROBUSTNESS

**Nils Philipp Walter**
CISPA Helmholtz Center
for Information Security
nils.walter@cispa.de

**Linara Adilova**
Research Center Trustworthy Data Science and Security
TU Dortmund
linara.adilova@tu-dortmund.de

**Jilles Vreeken**
CISPA Helmholtz Center
for Information Security
vreeken@cispa.de

**Michael Kamp**
Lamarr Institute, Technical University Dortmund &
Institute for AI in Medicine, University Hospital Essen
michael.kamp@uk-essen.de

## ABSTRACT

Despite their empirical success, neural networks remain vulnerable to small, adversarial perturbations. A longstanding hypothesis suggests that flat minima, regions of low curvature in the loss landscape, offer increased robustness. While intuitive, this connection has remained largely informal and incomplete. By rigorously formalizing the relationship, we show this intuition is only partially correct: flatness implies *local* but not *global* adversarial robustness. To arrive at this result, we first derive a closed-form expression for relative flatness in the penultimate layer, and then show we can use this to constrain the variation of the loss in input space. This allows us to formally analyze the adversarial robustness of the entire network. We then show that to maintain robustness beyond a local neighborhood, the loss needs to curve *sharply* away from the data manifold. We validate our theoretical predictions empirically across architectures and datasets, uncovering the geometric structure that governs adversarial vulnerability, and linking flatness to model confidence: adversarial examples often lie in large, flat regions where the model is confidently wrong. Our results challenge simplified views of flatness and provide a nuanced understanding of its role in robustness.

## 1 INTRODUCTION

Despite their success across a wide range of tasks, neural networks remain notoriously brittle under adversarial perturbations. Small, often imperceptible changes to the input can dramatically alter the prediction of a model. Understanding the structural properties that contribute to this vulnerability is central to building more robust systems. One property that has long attracted attention is the flatness of the loss surface. Earlier work suggested that flatter minima correlate with better generalization (Hochreiter & Schmidhuber, 1994; Jiang et al., 2019), however, the universality of this link remains an open question (Andriushchenko et al., 2023). Flatness also emerged as a potential indicator for adversarial robustness (Wu et al., 2020): a model whose loss landscape is locally flat in parameter space might resist small perturbations in input space. At first glance, this appears to be disconnected, since adversarial examples concern the change of the loss with respect to the input, while flatness quantifies the change with respect to the weights. Still, in simple settings, one can observe a connection. For a linear model and a given training sample $x$, we have

$$\ell(f(\mathbf{w}(x + \delta x)), y) = \ell(f(\mathbf{w}x + \mathbf{w}\delta x), y) = \ell(f((\mathbf{w} + \mathbf{w}\delta)x), y) = \ell(f(\hat{\mathbf{w}}x), y) \cong \ell(f(\mathbf{w}x), y)$$

The last step follows from the loss surface being flat. That is, by definition the loss does not change under small perturbations. This informal derivation suggests that input perturbations can, in simple cases, be reinterpreted as weight perturbations. However, it oversimplifies the nonlinear structure of neural networks. In this work, we resolve this disconnect through a formal theoretical analysis. The main challenge lies in accounting for nonlinearity, which fundamentally alters how perturbations

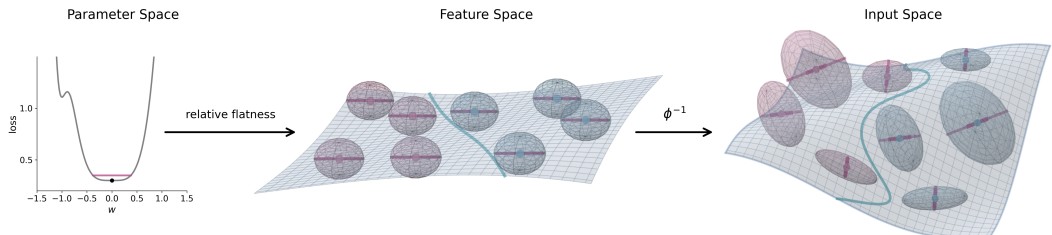

Figure 1: **Flatness shapes robustness through the network.** Flatness in parameter space (left) implies a minimum robustness radius in feature space (center), which maps back to warped but bounded regions in input space (right). This illustrates how local flatness in the weight space of the penultimate layer translates into localized but non-uniform robustness in the input space.

propagate through the network. To overcome this, we extend the theoretical results by Petzka et al. (2021), who related a carefully defined notion of flatness in a single layer, based on the trace of the hessian $Tr(H)$, that binds the robustness to perturbations in feature space. Here, we show that their theoretical framework can be extended to relate the flatness of the loss surface to adversarial robustness. Intuitively, we show that under theory-friendly conditions, flatness in feature space constrains how much the loss can change in input space. This relationship is illustrated in Figure 1: point-wise flatness, measured at individual inputs for a fixed parametrization, corresponds to spherical regions in feature space where the loss changes slowly. These map back to regions in input space, potentially warped or elliptical, where the loss also varies gradually, including under adversarial perturbations.

We then make this intuition precise. To this end, we derive a closed-form expression for relative flatness in the penultimate layer for the cross-entropy loss. This enables us to characterize how local curvature in parameter space influences the variation of the loss in feature space. By bounding the third derivative of the loss, we show that flatness at the penultimate layer induces a lower bound on the radius within which the loss remains approximately constant in feature space. Through the local Lipschitz continuity of the feature extractor, these spherical regions propagate back into the input space, where they become warped and anisotropic but retain their minimal safe diameter. Crucially, the size of these robust regions is dictated directly by the relative flatness at the penultimate layer.

Our derivations and experiments confirm a long-suspected relationship between flatness and robustness; yet they expose its limitations. The theoretical picture is appealing but incomplete. While it is tempting to believe that a flatter model is automatically more robust, our results show that this can be misleading. Flatness ensures local robustness around individual points where it is measured, but it does not characterize the structure of the loss landscape globally on all the data. In particular, we observe that first-order adversarial attacks tend to move examples from initially sharp regions into much flatter ones. That is, adversarial examples often settle in flat basins of the loss, which we call the *Uncanny Valley* (cf. Fig. 4). To understand this behavior, we revisit our closed-form expression for relative flatness in the penultimate layer. This expression reveals a tight connection between local geometry and model confidence: flatness tends to emerge in regions where the model is highly confident. In turn, this helps explain why adversarial examples can appear deceptively safe–despite being incorrect, they often reside in regions of low curvature and high confidence. This local flatness backpropagates through the network, influencing sensitivity in earlier layers. This theoretical lens enables us to rigorously invalidate overly simplified claims of the form: *Flat minima result in adversarially more robust networks*. In summary, we make the following contributions:

(i) We establish a theoretical connection between relative flatness and adversarial robustness and empirically validate these results.

(ii) We derive a closed-form expression for relative flatness in the penultimate layer, revealing its link to confidence and its influence on network sensitivity.

(iii) We introduce and analyze the *Uncanny Valley*, showing that adversarial examples often lie in flat, vast, high-confidence regions despite being misclassified.

These results provide a clearer understanding of how flatness and adversarial robustness interact. While flatness offers meaningful local guarantees, our theory reveals structural blind spots that still permit the existence of adversarial examples. This bridges parameter-space geometry and input-space robustness through both theoretical analysis and empirical evidence.

## 2 RELATED WORK

**Adversarial Examples**    Szegedy et al. (2014) introduced the notion of adversarial samples. An adversarial sample is a subtle perturbation of a benign input that remains imperceptible to humans but causes a model to make incorrect predictions. Numerous methods for crafting adversarial examples have been developed. Early approaches include the Fast Gradient Sign Method (FGSM) (Goodfellow et al., 2015), followed by more powerful and widely adopted techniques such as Projected Gradient Descent (PGD) (Madry et al., 2017) and the C&W (Carlini & Wagner, 2017). Further work expanded attack methods (Papernot et al., 2016; Kurakin et al., 2017; Narodytska & Kasiviswanathan, 2017; Brown et al., 2018; Alaifari et al., 2018; Andriushchenko et al., 2020; Croce & Hein, 2020; 2021), typically relying on gradient-based optimization. In general, all deep learning based models, including large language models are vulnerable to input manipulations (Li et al., 2020; Zou et al., 2023; Wei et al., 2024). Because of many established attacks and smoothness of the input space we here focus on the image domain.

**Flatness for Generalization & Robustness**    The notion of flat minima was introduced by Hochreiter & Schmidhuber (1994), who argued that such solutions correspond to simpler models with better generalization. This view gained empirical support from Keskar et al. (2017), who showed that small-batch training tends to converge to flatter minima with lower test error, while large-batch training leads to sharper, less generalizable solutions. Subsequent studies linked generalization to sharpness metrics such as the Hessian trace and maximum eigenvalue (Jiang et al., 2019; Neyshabur et al., 2017). These insights inspired a range of optimization techniques aimed at biasing training towards flatter regions of the loss surface. Methods like Entropy-SGD (Chaudhari et al., 2019), Stochastic Weight Averaging (SWA)(Izmailov et al., 2018), SWAD (Cha et al., 2021), and Sharpness-Aware Minimization (SAM) (Foret et al., 2021) explicitly or implicitly minimize sharpness and have been shown to improve generalization across various architectures and datasets.

The relation between flatness and generalization remains topic of debate, however. Dinh et al. (2017) showed that sharpness measured via the loss Hessian is not invariant under reparameterization, undermining raw curvature-based metrics. Kaur et al. (2022) and Andriushchenko et al. (2023) found that sharpness often reflects training hyperparameters more than generalization quality, and may not reliably predict out-of-distribution performance. To address these issues, reparameterization-invariant definitions such as adaptive sharpness (Kwon et al., 2021) and relative flatness (Petzka et al., 2021) were proposed, with the latter also providing theoretical generalization guarantees. Flatness has also been studied in the context of adversarial robustness. Stutz et al. (2021) analyzed curvature under $\ell_p$-bounded perturbations and observed that flatter input-space regions tend to correlate with robustness. Subsequent works showed that robust models often lie in flatter regions of the weight landscape (Zhao et al., 2020), and that sharpness-aware training alone can enhance robustness even without adversarial examples (Wu et al., 2020; Wei et al., 2023).

## 3 PRELIMINARIES

We assume a distribution $\mathcal{D}$ over an input space $\mathcal{X}$ and a label space $\mathcal{Y}$ with corresponding probability density function $P(X, Y) = P(Y \mid X)P(X)$. We consider models $f : \mathcal{X} \to \mathcal{Y}$ from a model class $\mathcal{F}$, and loss functions $\ell : \mathcal{Y} \times \mathcal{Y} \to \mathbb{R}_+$.

### 3.1 CLASSIC DEFINITION OF ADVERSARIAL EXAMPLES AND ROBUSTNESS

An adversarial example is a minimal perturbation $r^*$ added to a sample $x$ such that $\xi = x + r^*$ is misclassified. Formally, it is defined as an optimization problem.

**Definition 1** (Prediction-change adversarial example (Szegedy et al., 2014))**.** *Let* $f : \mathbb{R}^m \to \{1, \ldots, k\}$ *be a classifier,* $x \in [0, 1]^m$*, and* $l \in [k]$ *with* $l = f(x)$ *the predicted class. Then*

$$r^* = \arg \min_{r \in \mathbb{R}^m} \|r\|_2 \quad s.t. \quad f(x + r) \neq l \quad and \quad x + r \in [0, 1]^m,$$

*and the perturbed sample* $\xi = x + r^*$ *is called an **adversarial example**.*

Conversely, a classifier is said to be adversarially robust at a sample $x$ if its predictions remain constant under small perturbations, i.e., there exists no small perturbation such that the label flips.

**Definition 2** (Prediction-change pointwise robustness). *A classifier $f$ is said to be $\delta$-robust at $x \in \mathcal{X}$ if $\forall \xi \in B_\delta(x), \quad f(\xi) = f(x)$, where $B_\delta(x) = \{\xi \in \mathcal{X} \mid \|x - \xi\|_2 \leq \delta\}$ denotes the closed ball of radius $\delta$ around $x$.*

**Definition 3** (Prediction-change dataset robustness). *Given a dataset $S \subseteq \mathcal{X} \times \mathcal{Y}$, a classifier $f$ is $\delta$-robust over $S$ if it is $\delta$-robust at every $(x, y) \in S$ according to Definition 2.*

## 3.2 Loss-based Definitions of Adversarial Examples and Robustness

In this work, we connect the geometry of the *loss* surface with adversarial robustness. The classical definition of adversarial examples (Definition 1) only incorporates the loss implicitly through the prediction flip, making it unsuitable for directly analyzing how robustness relates to the structure of the loss landscape. To address this, we adopt a straight-forward modification.

**Definition 4** (Loss-change adversarial example ). *Let $f : \mathbb{R}^m \to \{1, \ldots, k\}$ be a classifier, $x \in [0, 1]^m$, and $l = f(x)$ the predicted class. Let $\ell : \{1, \ldots, k\} \times \{1, \ldots, k\} \to \mathbb{R}_+$ be a loss function, and let $\epsilon > 0$ be a threshold. Then*

$$r^* = \arg \min_{r \in \mathbb{R}^m} \|r\|_2 \quad s.t. \quad \ell(f(x+r), l) - \ell(f(x), l) > \epsilon \quad and \quad x + r \in [0, 1]^m,$$

*and the perturbed sample $\xi = x + r^*$ is called a $(\delta, \epsilon)$-adversarial example, where $\delta = \|r\|_2$.*

Instead of requiring a change in prediction, we require the loss to increase by more than a threshold $\epsilon$. This retains the core idea of adversarial examples i.e. how small input perturbations have a malicious effect on the model, while capturing how perturbations in the input space translate to changes in the loss surface, rather than relying solely on discrete prediction flips. This definition provides a generalization of the classic definition. Any adversarial example from Definition 1 satisfies Definition 4 for some $\epsilon > 0$, and by using a conservative $\epsilon > \log(k)$ for cross-entropy loss smaller than $\ln 2$, we can ensure a prediction-flip, which recovers the classical criterion. We make this precise in Lemma 10. Analogous to Definition 2, we define robustness against adversarial loss changes. Note that in the low-confidence regime (loss $> \ln 2$), flatness is not informative on adversarial robustness, since predictions can flip without changing the loss.

**Definition 5** (Loss-change pointwise robustness). *Let $f : \mathcal{X} \to \mathcal{Y}$ be a classifier, $\ell : \mathcal{Y} \times \mathcal{Y} \to \mathbb{R}_+$ a loss function, and $(x, y) \in \mathcal{X} \times \mathcal{Y}$ a sample with $\ell(f(x), y) < \ln 2$. Given thresholds $\delta > 0$ (perturbation radius) and $\epsilon > 0$ (loss tolerance), we say that $f$ is $(\delta, \epsilon)$-robust at $(x, y)$ if $\quad \forall \xi \in B_\delta(x), \ell(f(\xi), y) - \ell(f(x), y) \leq \epsilon$.*

In this setting, robustness requires that small perturbations do not substantially increase the loss. This makes robustness a continuous property, where the degree of robustness is controlled by the choice of $\epsilon$. A smaller $\epsilon$ enforces stricter stability, while a larger one allows more tolerance. For example, a perturbation may not flip the predicted label but can still reduce the model's confidence, leading to a higher loss. The classical definition overlooks such cases, but the loss-based view captures such vulnerabilities naturally. We now extend the pointwise definition to the dataset level.

**Definition 6** (Loss-change dataset robustness). *Given a dataset $S \subseteq \mathcal{X} \times \mathcal{Y}$, a classifier $f$ is $(\delta, \epsilon)$-robust over $S$ if it is $(\delta, \epsilon)$-robust at every $(x, y) \in S$ according to Definition 5.*

It is easy to see that by choosing $\epsilon_*(S; \delta) := \min_{(x,y) \in S} \inf_{\xi \in B_\delta(x): f(\xi) \neq y} [\ell(f(\xi), y) - \ell(f(x), y)]$, one recovers the prediction-flip criterion. This expression captures the smallest loss increase over all $\delta$-perturbations that change the prediction. With this choice, no prediction flip will incur. This loss-based formulation is strictly stronger, though, as it requires small loss under perturbation even for high-confidence predictions—enforcing a stricter, uniform standard across the dataset. Alternatively, allowing $\epsilon$ to vary per sample recovers the original pointwise definition exactly and offers greater flexibility. For ease of analysis, we focus on the stronger version with a fixed global $\epsilon$.

## 4 What is Flatness Made Of?

Flatness measures aim to quantify how sensitive a neural network's loss is to perturbations in parameters. Empirical approaches estimate flatness heuristically by applying random perturbations to model weights and observing the resulting loss variations (Keskar et al., 2017). Analytical methods quantify curvature (Hochreiter & Schmidhuber, 1994), using the trace or eigenvalue spectrum of

the Hessian of the loss. Both approaches, however, are expensive to compute in high-dimensional settings and offer limited support for analytical reasoning—especially eigenvalue-based measures, which are harder to manipulate symbolically and less amenable to theoretical analysis.

Among Hessian-based metrics, the trace is more practical and often used in theoretical analysis due to its simpler structure. However, the raw trace is not invariant to reparameterization and thus does not reliably correlate with generalization (Dinh et al., 2017). To address these limitations, Petzka et al. (2021) proposed a reparameterization-invariant flatness measure, *relative flatness*, derived from a theoretical decomposition of the generalization gap. The measure combines the trace of the Hessian with the norm of the weights in a single layer. They show that, under mild technical assumptions, low relative flatness implies a small generalization gap. Importantly, they informally argue that it is sufficient to compute this quantity at any individual layer, in particular at the *penultimate layer*, avoiding the need to analyze the full network. Below, we provide a formal justification for this claim. Empirically, relative flatness also correlates strongly with generalization performance, making it both theoretically grounded and practically informative. Formally, given a network decomposed into a feature extractor $\phi : \mathcal{X} \to \mathbb{R}^d$ mapping to the penultimate layer $\mathbf{w} \in \mathbb{R}^{k \times d}$ and a classifier $g$ applying softmax, i.e., $f(x) = g(\mathbf{w}\phi(x))$ relative sharpness is defined as,

$$\kappa_{Tr}(\mathbf{w}) := \|w\|_2 Tr(H(\mathbf{w}, S)) \ , \tag{1}$$

where $Tr$ denotes the trace, and $H$ is the Hessian of the loss computed on dataset $S$ wrt $\mathbf{w}$:

$$H(\mathbf{w}, S) = \frac{1}{|S|} \sum_{(x,y) \in S} \left( \frac{\partial^2}{\partial w_i \partial w_j} \ell \left( g(\mathbf{w}\phi(x)), y \right) \right)_{i,j \in [km]} \ . \tag{2}$$

This simplified measure (Petzka et al., 2019) upper bounds full relative flatness (Lemma A.2 Han et al., 2025). Counter-intuitively, this measure is small when the loss surface is flat, since in that case the trace of the Hessian is small. Similarly, a large value of $\kappa_{Tr}(\mathbf{w})$ indicates that the loss surface is sharp. To avoid confusion, we therefore call $\kappa_{Tr}(\mathbf{w})$ *relative sharpness*.

To understand the geometric structure of relative sharpness and enable the theoretical analysis in Section 4, we need an explicit formula for the Hessian. We therefore derive a closed-form expression of the Hessian for the cross-entropy loss, which reveals the underlying geometry and serves as the foundation for our theoretical analysis. The full derivation is provided in Appendix C.4. For a single example $(x, y)$, defining the softmax predictions as $\hat{y} = g(\mathbf{w}\phi(x))$, our derived expression is

$$H(\mathbf{w}, \{(x,y)\}) = (\text{diag}(\hat{y}) - \hat{y}\hat{y}^T) \otimes \phi\phi^T \tag{3}$$

yielding a simple analytical formula,

$$\kappa_{Tr}(\mathbf{w}) = \|w\|_2 Tr(H(\mathbf{w}, \{(x,y)\})) = \|w\|_2 \sum_{j=1}^{k} \hat{y}_j (1 - \hat{y}_j) \sum_{i=1}^{d} \phi_i^2 \ , \tag{4}$$

Thus, the geometry as measured by relative sharpness consists of three intuitive components: the network's confidence in terms of softmax output (captured by $\hat{y}_j(1 - \hat{y}_j)$), the scale of the feature representation ($\phi_i^2$), and the weight magnitude ($\|w\|_2$). Although this derivation only applies to the penultimate layer, we will demonstrate that this geometry propagates to earlier layers. This analytical form is key to bounding adversarial susceptibility using relative sharpness in Section 5.

**Why the last layer is sufficient** For any recitified affine neural network composed of linear, convolutional and BatchNorm layers, curvature propagates cleanly through the architecture. Let $f_l(x) = \text{ReLU}(W_l x)$ denote the $l$-th layer, with pre-activations $a^l = W_l x^{l-1}$. Define the gradient and Hessian of the loss $\ell$ with respect to these pre-activations as $g^l = \partial\ell/\partial a^l$ and $H^l = \partial^2\ell/\partial(a^l)^2$. Since ReLU has zero second derivative almost everywhere, and each affine map introduces no additional curvature, the chain rule yields the recurrence

$$H^{l-1} = W_l^\top D^l H^l D^l W_l,$$

where $D^l = \text{diag}(1_{a^l > 0})$ masks the active units. The Hessian block with respect to the weights of layer $l$ takes the Kronecker-factored form

$$H_{W_l} = (x^{l-1}x^{l-1\top}) \otimes (D^l H^l D^l).$$

If the final-layer Hessian vanishes, so do all earlier activation- and weight-space blocks. Therefore, the curvature profile of the entire network is fully determined by the last layer, and any sharpness measure based on the Hessian attains its maximum there. We formalize this result in Appendix C.4.

This structure reveals that all curvature in hidden layers is inherited from the output layer. In particular, if $H^L = 0$, then by induction all downstream Hessians also vanish. Even when $H^L \neq 0$, the conditioning of every earlier block is shaped by it. As a result, any sharpness measure based on the Hessian is strongly influenced by the final layer and its immediate input, justifying our focus on the penultimate layer in what follows. We formalize this result in Appendix B. [1]

## 5 Connecting Relative Sharpness to Adversarial Robustness

In this section we make explicit how relative sharpness influences adversarial robustness. To achieve this, we first connect input perturbations to changes in the output, then relate these changes in feature space to parameter perturbations, i.e., relative sharpness. Second, we use Taylor expansion around adversarial examples to bound the loss increase under such perturbations. Finally, we derive the robustness bound that reveals the role of relative sharpness. This precisely shows how relative sharpness influences the relationship between input perturbations and loss variations, though the resulting bound is not intended as practically tight guarantees. As the derivations are technical, we present the key ideas and proof sketches here and defer full details to the appendix.

**Connecting input to feature space** We first formalize how input-space perturbations propagate into feature-space representations. This is essential since neural networks transform perturbations nonlinearly, complicating the input-to-parameter relationship. Formally, consider a model $f(x) = g(\mathbf{w}\phi(x))$ with feature extractor $\phi$, classifier $g$, and weight matrix $\mathbf{w}$. Such a model nonlinearly relates input perturbations to those in feature space (Petzka et al., 2021). Specifically, we represent the adversarial representation $\phi(\xi)$ as a perturbation of the clean representation $\phi(x)$.

**Lemma 7.** *Let $f = g(\mathbf{w}\phi(x))$ be a model with $\phi$ L-Lipschitz and $\|\phi(x)\| \geq r$, and $\xi, x \in \mathcal{X}$ with $\|\xi - x\| \leq \delta$, then there exists a $\Delta > 0$ with $\Delta \leq L\delta r^{-1}$, such that $\phi(\xi) = \phi(x) + \Delta A\phi(x)$, where $A$ is an orthogonal matrix.*

The proof is provided in Appx. C.1. Thus we can relate the adversarial example $\xi$ to perturbations in the weights $\mathbf{w}$ of the representation layer, for which we use the linearity argument from Petzka et al. (2021). Since we can express $\phi(\xi)$ as $\phi(x) + \Delta A\phi(x)$, we can then use the linearity of the representation layer to relate the perturbation of input to a perturbation in weights. That is,

$$\ell(f(\xi), y) = \ell(g(\mathbf{w}\phi(\xi)), y) = \ell(g(\mathbf{w}(\phi(x) + \Delta A\phi(x))), y) = \ell(g((\mathbf{w} + \Delta\mathbf{w}A)\phi(x), y) .$$

This means that we can express an adversarial example as a suitable perturbation of the weights $\mathbf{w}$ of the representation layer, where the magnitude is bounded by $\Delta \leq L\delta r^{-1}$.

**Bounding loss increase** Next, we bound the loss difference between $\xi$ and $x$. For ease of notation, we define $\ell(\mathbf{w} + \Delta\mathbf{w}A) := \ell(g((\mathbf{w} + \Delta\mathbf{w}A)\phi(x), y)$. The Taylor expansion of $\ell(\mathbf{w} + \Delta\mathbf{w}A)$ at $\mathbf{w}$ gives the following

$$\ell(\mathbf{w} + \Delta\mathbf{w}) = \ell(\mathbf{w}) + \langle \Delta\mathbf{w}A, \nabla_\mathbf{w}\ell(\mathbf{w})\rangle + \frac{\Delta^2}{2}\langle \mathbf{w}A, H\ell(\mathbf{w})(\mathbf{w}A)\rangle + R_2(\mathbf{w}, \Delta) ,$$

where $H\ell(\mathbf{w})$ is the Hessian of $\ell(\mathbf{w})$. If we now maximize over all $A$ with $\|A\| \leq 1$, it follows that $\langle \mathbf{w}A, H\ell(\mathbf{w})(\mathbf{w}A)\rangle \leq \|\mathbf{w}\|_F^2 Tr(H\ell(\mathbf{w})) = \kappa_{Tr}^\phi(\mathbf{w})$. Therefore, we have

$$|\ell(f(\xi), y) - \ell(f(x), y)| \leq \Delta\|\mathbf{w}\|_F\|\nabla_\mathbf{w}\ell(\mathbf{w})\|_F + \frac{\Delta^2}{2}\kappa_{Tr}^\phi(\mathbf{w}) + R_2(\mathbf{w}, \Delta) . \tag{5}$$

The remainder depends on the partial third derivatives of the loss. We show in Appdx. C.2 that for feature extractor $\phi$ that is $L$-Lipschitz, it can be bound by $4^{-1}kmL^3$. With this we can bound the difference between the loss suffered on an adversarial example $\xi$ and the loss on a clean example $x$ for a converged model as follows.

---

[1] This does not hold for transformer since attention can introduce curvature.

**Proposition 8.** *For $(x, y) \in \mathcal{X} \times \mathcal{Y}$ with $\|x\| \leq 1$ for all $x \in \mathcal{X}$, a model $f(x) = g(\mathbf{w}\phi(x))$ at a minimum $\mathbf{w} \in \mathbb{R}^{m \times k}$ with $\phi$ L-Lipschitz and $\|\phi(x)\| \geq r$, and the cross-entropy loss $\ell(\mathbf{w}) = \ell(g(\mathbf{w}\phi(x)), y)$ of $f$ on $(x, y)$, it holds for all $\xi \in \mathcal{X}$ with $\|x - \xi\|_2 \leq \delta$ that*

$$\ell(f(\xi), y) - \ell(f(x), y) \leq \frac{\delta^2}{2r^2} L^2 \kappa_{Tr}^\phi(\mathbf{w}) + \frac{\delta^3}{24r^3} kmL^6 \ .$$

We defer the proof to Appx. C.2. This analytical result clarifies explicitly how relative sharpness controls loss sensitivity to perturbations.

**Bounding adversarial robustness** We now derive an explicit expression for the maximum perturbation radius $\delta$ within which the loss increase remains bounded by a given threshold $\epsilon$. Solving the Taylor-based bound for $\delta$ yields a cubic equation with both linear and quadratic terms. While this equation does not admit a clean closed-form solution in general, we isolate and present the dominant terms to illustrate the key dependencies:

**Proposition 9.** *[Informal] For a dataset $S \subset \mathcal{X} \times \mathcal{Y}$ with $\|x\| \leq 1$ for all $x \in \mathcal{X}$, a model $f(x) = g(\mathbf{w}\phi(x))$ at a minimum $\mathbf{w} \in \mathbb{R}^{m \times k}$ wrt. $S$, with $\phi$ L-Lipschitz and $\|\phi(x)\| \geq r$, and the cross-entropy loss $\ell(\mathbf{w}) = \ell(g(\mathbf{w}\phi(x)), y)$ of $f$ on $(x, y)$, $d$ being the L2-distance, and $\epsilon > 0$, $f$ is $(\epsilon, \delta, S)$-robust against adversarial examples with*

$$\delta \ \propto \ \frac{\epsilon^{\frac{1}{3}}}{L \, \kappa_{Tr}(\mathbf{w})^{\frac{1}{3}}} \ + \ \frac{r \, k \, m \, L^2}{\kappa_{Tr}(\mathbf{w})} \qquad \lim_{\kappa_{Tr}(\mathbf{w}) \to 0} \delta = \frac{r}{L} \left( \frac{24 \, \epsilon}{kmL^3} \right)^{\frac{1}{3}}$$

*Hence decreasing sharpness enlarges the certified radius, yet the bound* does not blow up*; it saturates at the value above.*

The formal version and full proof are provided in Appendix C.3. Proposition 9 precisely reveals how relative sharpness constrains the extent to which the loss can change under input perturbations. Beyond its analytical role, it aligns with geometric intuition: flatter models in the sense of lower $\kappa_{Tr}(\mathbf{w})$, together with smooth feature maps, exhibit locally stable behavior. Interestingly, the explicit formula for relative sharpness in Equation 4 reveals that sharpness *decreases* as model confidence *increases*. When the predicted class probability approaches one, the corresponding term $\hat{y}_j(1 - \hat{y}_j)$ vanishes, reducing the overall sharpness. This implies that highly confident predictions are, at least *locally*, more robust to input perturbations than uncertain ones. This stands in contrast to the common claim that high-confidence predictions are cause for adversarial susceptibility. Instead, our analysis suggests that high-confidence regions may actually promote local stability by flattening the loss surface. We revisit and empirically validate this observation in Section 6.

## 6 FROM THEORY TO PRACTICE

In this section, we empirically investigate how relative sharpness influences adversarial robustness. To test the theoretical predictions from Section 4, we exploit a key insight: by scaling the penultimate layer weights $\mathbf{w}$ by a factor $s$ i.e. $\mathbf{w}_s = s\mathbf{w}$, we can directly control the loss surface curvature without retraining. Smaller scaling factors yield sharper networks, while larger values produce flatter ones. This method allows us to precisely control sharpness in a systematic and reproducible manner, in contrast to SAM-based training approaches.

We design three experiments to examine different aspects of the sharpness-robustness connection. First, we examine whether the curvature properties at the penultimate layer propagate throughout the network, validating our derivations in Section 4 and our focus on this layer as representative of the global loss landscape. Second, we test our theoretical insight from Proposition 9 by examining whether networks with reduced sharpness demonstrate smaller loss changes when subjected to adversarial perturbations, directly validating whether flatter networks exhibit improved stability under input perturbations. Additionally, we show that adversarial training significantly increases the guaranteeable radius. Third, we investigate potential limitations of using relative sharpness as a global robustness and correctness indicator given its close connection to model confidence.

**General setting.** We evaluate our theoretical predictions on standard architectures and datasets. We train ResNet-18 (He et al., 2016), WideResNet-28-4 (Zagoruyko & Komodakis, 2016),

DenseNet-121 (Huang et al., 2017), and VGG-11 with BatchNorm (Simonyan & Zisserman, 2014) on both CIFAR-10 and CIFAR-100. All models are trained using stochastic gradient descent for 100 epochs with an initial learning rate of 0.1, cosine learning rate scheduling, and weight decay of $10^{-4}$. For clarity of presentation, we focus our main analysis on ResNet-18 trained on CIFAR-10. Our results are consistent across all architectures and datasets; hence we report on this standard combination. Complete results for other architectures and CIFAR-100 are provided in Appendix D.

## 6.1 SHARPNESS PREDICTS LOSS SENSITIVITY UNDER ADVERSARIAL PERTURBATIONS

We now empirically test our core claim that relative sharpness governs local loss sensitivity and shapes adversarial robustness. By scaling the network post-hoc, we isolate the effect of sharpness while keeping all model parameters fixed, enabling controlled investigation of its role. While flattening the network does not eliminate adversarial examples entirely, it significantly affects how rapidly the loss increases in their vicinity. Specifically, we examine how local sharpness influences the magnitude of loss increase as defined in Definition 4.

**Setup.** To this end, we generate adversarial examples using a weaker attack PGD-$\ell_2$ (25 steps, $\epsilon = 0.025$, step size $\alpha = 0.001$). This attack incrementally increases perturbations by $0.001\ell_2$ per iteration, achieving a robust test accuracy of 90.33% on the original model. We evaluate scaled networks with scaling factors $s \in \{0.25, 0.5, 1, 2.5, 5, 10, 50\}$ along the trajectories of these attacks, measuring how the loss evolves. Results are summarized in Figure 2, with a subset of scaling factors presented for clarity. Comprehensive results for all scaling values are provided in the appendix.

**Loss Increase Analysis.** Figure 2a shows the histogram of loss increases between clean examples $x_0$ and final adversarial iterates $x_{25}$. As predicted by Equation 5, increasing the scale factor (flattening the network) substantially reduces observed loss increases, approaching zero for large scaling values. This occurs because flattening creates locally stable regions–or basins–around inputs.

**Sharpness Creates Basins.** Figure 2b illustrates this phenomenon for a representative example. As we increase the sclaing $s$, the loss remains nearly flat over progressively larger distances along the attack trajectory. For small scaling factors (e.g., $s = 0.5$), the loss increases steadily from the first iteration, while for large factors (e.g., $s = 50$), the loss stays almost constant throughout the entire trajectory until a sharp *take-off point* where it suddenly strongly increases, which marks the boundary of the basin. Specifically, flatter loss surfaces, as derived in Proposition 9, can progressively guarantee larger adversarial robustness radii $\delta$ for given $\epsilon$, but only up to the take-off point, which represents the effective limit of sharpness when exclusively enforced by reducing $tr(H)$.

**Per Sample Basin.** The location of the take-off point varies across samples, as shown in Figure 2c. In other words, the width of the basin is sample-dependent, and consequently, so is the guaranteeable radius. Following conventional wisdom in the field, we observe that basin width strongly correlates with the sharpness measured at the clean input–specifically, flatter networks (lower sharpness) yield broader basins. This relationship is empirically confirmed in Figure 2d. This observation aligns with the established intuition that reduced sharpness increases robustness, though within a radius that remains insufficient to prevent label flips.

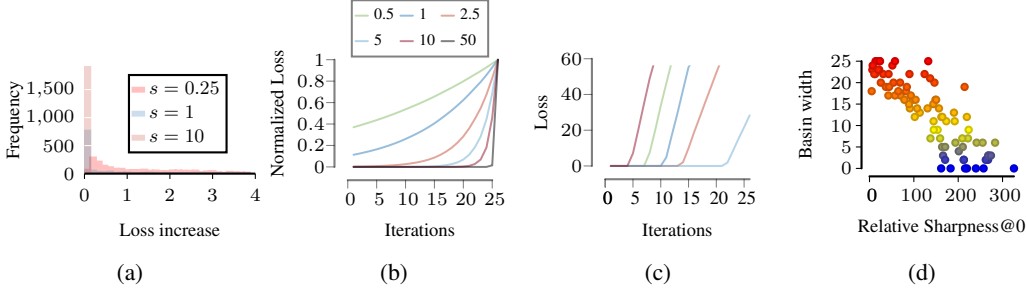

Figure 2: *Loss geometry.* **(a)** We report the distribution of the loss increase for varying scaling values. **(b)** We show for one example how the basin formes as we increase the scaling. We use the normalized loss to plot all in one axis. **(c)** We show examples of samples exhibiting different basin widths. **(d)** We plot the sharpness measured at the test data ($s = 1$) and the basin width.

## 6.2 Adversarial Training Increases Basin Width

We also run the same evaluation as in Section 6.1 with an adversarially trained ResNet. For training, we use standard adversarial training with PGD-$\ell_\infty$ (10 steps, $\epsilon = 8/255$, step size $\alpha = 2/255$). Given that this will result in a more adversarially robust model, we use a stronger attack as in the previous section, namely PGD-$\ell_2$ (50 steps, $\epsilon = 0.5$, step size $\alpha = 0.01$). This attack explores a radius $20\times$ bigger radius than compared to the weaker attack. We present the results in Figure 3.

In general, we observe trends that are consistent with all other models and datasets. Specifically, as we scale the model, the loss increase diminishes, and we observe a similar scaling pattern and correlation between sharpness@0 and basin width. The primary distinction is that the basin width is significantly larger (approx. $20\times$) because we measure distance in iterations. Consequently, the classification remains constant within a larger radius, even though the local curvature measured by $Tr(H)$ remains comparable (e.g., 0.6 clean vs. 1.0 adversarial). As shown in Equation 4, the small disparity stems from the fact that adversarially trained networks are significantly less confident.

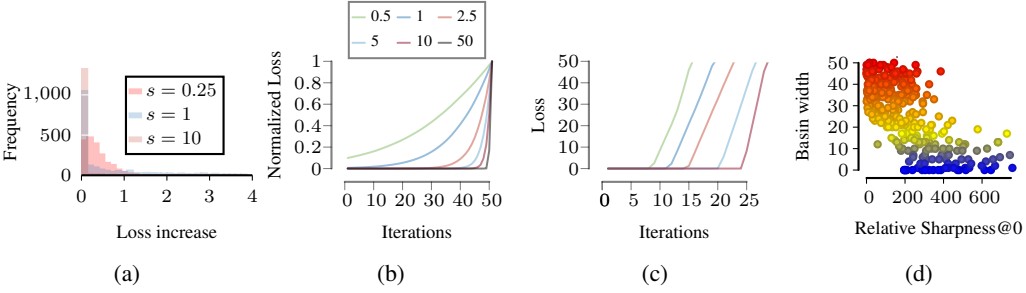

(a)      (b)      (c)      (d)

Figure 3: *Loss geometry for adversarially trained* RESNET-18 *(trained via PGD-$\ell_\infty$, $\epsilon = 8/255$) evaluated under a PGD-$\ell_2$ attack ($\epsilon = 0.5$, $\alpha = 0.01$).* **(a)** We report the distribution of the loss increase for varying scaling values. **(b)** We show for one example how the basin forms as we increase the scaling. We use the normalized loss to plot all on one axis. **(c)** We show examples of samples exhibiting different basin widths. **(d)** We plot the sharpness measured at the test data ($s = 1$).

## 6.3 Sharpness Can Be Deceiving

In the previous experiments, we validated our theoretical derivations showing that relative sharpness governs local loss sensitivity, with flatter networks exhibiting limited loss variation under perturbations. We now turn to a more nuanced question. The flatness literature carries an implicit hope that geometric properties can be linked to correctness–that flatter minima naturally correspond to regions where the model generalizes better and makes accurate predictions. This hope is the implicit motivation of seeking flatter minima. To investigate this, we measure relative sharpness (the same holds for common classification losses, see Lemma 11 & Lemma 12) along PGD-$\ell_\infty$ attack trajectories (10 steps, $\epsilon = 8/255$, $\alpha = 2/255$) to examine the evolving geometry.

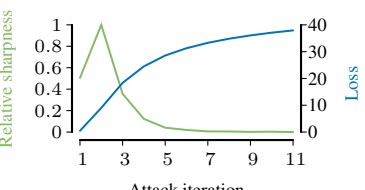

Figure 4: *Uncanny Valley.* On the PGD-trajectory, sharpness quickly peaks and then decreases to zero, while loss keeps increasing.

We plot the results in Fig 4, which reveals a striking phenomenon: as the adversarial attack progresses, sharpness initially increases as expected. It then, however, dramatically decreases to near-zero values while the loss plateaus. This demonstrates that adversarial examples settle in exceptionally flat regions. Our closed-form expression for relative sharpness explains this *uncanny valley*; in high-confidence regions, confidence dominates the geometry, creating flat valleys where gradients nearly vanish. This challenges the hope that flatness indicates correctness–these regions are geometrically stable yet correspond to confident misclassifications. The sharp peak in relative sharpness marks the decision boundary, where model uncertainty creates maximum curvature. The adversarial attack traverses this ridge before descending into flat, high-confidence regions on the wrong side. This reveals how confidence shapes the loss surface geometry—sharp at decision boundaries, flat in confident regions—*regardless* of correctness.

## 7  DISCUSSION & CONCLUSION

In this work, we formalized the long-suspected connection between flatness and adversarial robustness, revealing both the promise and fundamental limitations of sharpness as a mediator for robustness. Our analysis bridges parameter-space and input-space perspectives through rigorous theoretical derivations and extensive empirical validation. Traditional prediction-flip definitions of adversarial examples proved insufficient for this analysis, so we introduced a loss-based formulation (Definition 4) that enables direct analysis of how perturbations propagate through the loss landscape. This approach connects the geometric properties at the penultimate layer to adversarial vulnerability in input space. We extended (Petzka et al., 2021) framework to the adversarial setting, deriving explicit bounds (Proposition 9) that show how relative flatness constrains the adversarial perturbation radius on the points it is measured on. The robustness guarantee scales as $\epsilon^{1/3}/\kappa_{Tr}(w)^{1/3}$, providing a rigorous characterization of parameter-space geometry influences adversarial robustness.

Our empirical investigations confirmed these theoretical predictions while exposing the myopic nature of flatness-based robustness. Through controlled experiments, we showed that while flatter networks exhibit larger stability basins around individual inputs, these basins remain insufficient for practical adversarial robustness. The basins have finite width with sharp boundaries that act as "slingshots," rapidly increasing the loss once crossed. We observe a string phenomenon; starting from relatively flat regions, attacks traverse increasingly sharp areas before descending into exceptionally flat valleys. The peak corresponds to the decision boundary where predictions flip i.e maximal uncertainty. Beyond lie broad broad flat regions, termed the Uncanny Valley, where models are maximally confident yet wrong. This shows that confidence manifests in the geometry of the loss surface and that sharpness can be deceiving.

Our decision to analyze the network in two parts—feature extractor and classifier—proved crucial for understanding the true nature of flatness in neural networks. By focusing our theoretical analysis on the penultimate layer, we could isolate how classifier confidence dominates standard flatness measures. This decomposition revealed that $Tr(H)$-based metrics conflate two different properties: the geometric stability of learned representations and the confidence of the final classifier. When measuring flatness over the entire network, high confidence in the classifier can mask the true sensitivity of the feature extractor. Our approach demonstrates that meaningful robustness evaluation requires separately assessing these components. While the penultimate layer analysis might initially seem like a limitation, it actually enabled us to uncover how confidence effects propagate through the network and shape the global loss landscape.

In conclusion, by extending relative flatness to the adversarial setting, we have derived explicit bounds that confirm flatness does influence robustness—but only through local mechanisms that cannot extend globally. Our work demonstrates that flatness provides robustness within local basins whose size is fundamentally constrained by the confidence-flatness coupling. As long as flatness measures at the penultimate layer are dominated by the confidence term $\hat{y}_j(1 - \hat{y}_j)$, they cannot distinguish between true geometric robustness and mere confident predictions. This helps explain why optimizing for flatness, while beneficial, has proven insufficient for achieving strong adversarial robustness. While $Tr(H)$-based flatness works as a local robustness indicator, achieving global robustness requires developing new notions of flatness that meaningfully separate the contributions of the feature extractor from classifier confidence.

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

APPENDIX

## A  GENERALIZED DEFINITION OF ADVERSARIAL EXAMPLES

**Lemma 10.** *Let $f : \mathcal{X} \to \mathcal{Y}$ be a classifier, $l : \mathcal{Y} \times \mathcal{Y} \to \mathbb{R}_+$ a loss function (e.g., cross-entropy), and $(x, y) \in \mathcal{X} \times \mathcal{Y}$ a sample with $f(x) = y$. Suppose $\xi \in B_\delta(x)$ is a classical adversarial example, i.e., $f(\xi) \neq y$. Then there exists $\epsilon > 0$ such that*

$$l(f(\xi), y) - l(f(x), y) > \epsilon,$$

*i.e., $\xi$ is an adversarial loss change according to Definition 4.*

*Conversely, if $l$ satisfies a separation condition such that for all incorrect predictions $f(\xi) \neq y$ we have*

$$l(f(\xi), y) \geq \log(k),$$

*then any adversarial loss change with threshold $\epsilon > \log(k)$ implies $f(\xi) \neq y$.*

*Proof.* Since $f(x) = y$, we have that the loss at $x$ is small, typically $l(f(x), y) \approx 0$ for standard losses like cross-entropy. If $\xi$ satisfies $f(\xi) \neq y$, then under standard classification losses, $l(f(\xi), y)$ is large, typically $l(f(\xi), y) \gg 0$. Thus, the difference

$$l(f(\xi), y) - l(f(x), y) \approx l(f(\xi), y)$$

is positive and large. Choosing any $\epsilon$ satisfying $0 < \epsilon < l(f(\xi), y) - l(f(x), y)$ ensures the loss-change condition is satisfied.

Conversely, assume that for incorrect predictions $f(\xi) \neq y$, we have $l(f(\xi), y) \geq \log(k)$, as is the case under cross-entropy loss when the model assigns uniform or lower probability to the correct class. If $l(f(\xi), y) - l(f(x), y) > \epsilon$ for some $\epsilon > \log(k)$, then necessarily $l(f(\xi), y) > \epsilon > \log(k)$, implying that $f(\xi) \neq y$, because correct predictions would have loss much smaller than $\log(k)$. $\square$

## B  THE PENULTIMATE LAYER SHAPES THE HESSIAN

We consider a feedforward neural network of the form $\ell(f_L \circ f_{L-1} \circ \cdots \circ f_1(x))$, where each layer is defined recursively by $f_l(u) = \mathrm{ReLU}(W_l u)$. For each layer $l$, we define the pre-activation $a^l = W_l x^{l-1}$ and the post-activation $x^l = \mathrm{ReLU}(a^l)$. Let $g^l = \partial \ell / \partial a^l \in \mathbb{R}^{m_l}$ and $H^l = \partial^2 \ell / \partial (a^l)^2 \in \mathbb{R}^{m_l \times m_l}$ denote the gradient and Hessian with respect to the pre-activations. We define the binary diagonal gating matrix $D^l = \mathrm{diag}(1_{a^l > 0})$, which captures the ReLU derivative.

To compute derivatives efficiently through the network, we apply the second-order chain rule for scalar-valued functions. For a composition $y = g(u(x))$, where $x \in \mathbb{R}^n$, $u(x) \in \mathbb{R}^m$, and $y \in \mathbb{R}$, the Hessian satisfies

$$H_x = J_u^T H_y J_u + \sum_k \frac{\partial y}{\partial u_k} H_{u,k},$$

where $J_u = \partial u / \partial x$ and $H_{u,k} = \partial^2 u_k / \partial x^2$. Applying this to the ReLU network, we observe that all affine transformations $a^l = W_l x^{l-1}$ are linear in their input, so the second-order term $\sum_k g_k H_{u,k}$ vanishes. Similarly, the nonlinearity $f(u) = \mathrm{ReLU}(u)$ has second derivative zero almost everywhere, so no curvature is introduced at the nonlinearity either.

Under these conditions, the gradient and Hessian propagate layer by layer as

$$g^{l-1} = W_l^T D^l g^l, \qquad H^{l-1} = W_l^T D^l H^l D^l W_l.$$

This recurrence propagates curvature information purely through the affine structure and the ReLU activation mask. To compute the exact Hessian with respect to the weights of layer $l$, we vectorize the weight matrix $W_l$ columnwise as $w_l = \mathrm{vec}(W_l) \in \mathbb{R}^{m_l n_{l-1}}$. Differentiating the affine map $a^l = W_l x^{l-1}$ with respect to $w_l$ yields the Jacobian $J_l = x^{l-1} \otimes I_{m_l}$. Applying the second-order chain rule with respect to $w_l$ then yields the exact weight Hessian block

$$H_{W_l} = J_l H^l J_l^T = (x^{l-1} x^{l-1T}) \otimes (D^l H^l D^l).$$

The full backward recursion thus proceeds as follows. Starting from the output layer, where $g^L$ and $H^L$ are obtained analytically from the loss function, we iterate backward for $l = L, L-1, \ldots, 1$, computing at each layer the gradient $g^{l-1}$, the activation-space Hessian $H^{l-1}$, and the weight-space Hessian block $H_{W_l}$ as

$$g^{l-1} = W_l^T D^l g^l, \qquad H^{l-1} = W_l^T D^l H^l D^l W_l, \qquad H_{W_l} = (x^{l-1} x^{l-1T}) \otimes (D^l H^l D^l).$$

This derivation follows the second-order backpropagation framework originally introduced by Becker & LeCun (1989) and Bishop (1992). The Kronecker-factored form of the weight-space Hessian is the foundation for scalable second-order optimisers such as K-FAC Martens & Grosse (2015). In the ReLU case, the absence of second-order contributions from the nonlinearity implies that all curvature originates at the loss layer and propagates backward linearly through the active part of the network.

Let $H^L = \partial^2 \ell / \partial (a^L)^2$ denote the Hessian with respect to the pre-activations at the final layer. At a differentiable local minimum, the gradient vanishes and the Hessian is positive-semidefinite: $H^L \succeq 0$. In particular, all eigenvalues $\lambda_i$ of $H^L$ are non-negative. If in addition the trace vanishes, $\mathrm{tr}(H^L) = 0$, then the sum of the eigenvalues is zero. Since each $\lambda_i \geq 0$, it follows that $\lambda_i = 0$ for all $i$.

We now consider the spectral decomposition of $H^L$, which exists because the matrix is symmetric:

$$H^L = Q \Lambda Q^T, \qquad \Lambda = \mathrm{diag}(\lambda_1, \ldots, \lambda_n),$$

where $Q$ is orthogonal and $\Lambda$ contains the eigenvalues of $H^L$. As shown above, $\Lambda = 0$, so the decomposition becomes

$$H^L = Q\,0\,Q^T = 0.$$

Hence, if $H^L \succeq 0$ and $\mathrm{tr}(H^L) = 0$, then $H^L = 0$.

For a ReLU hidden layer the second derivative of the activation vanishes almost everywhere, so the curvature transfer matrix reduces to $B_l = D^l H^l D^l$ with $D^l = \mathrm{diag}(1_{a^l > 0})$. The second-order back-propagation recurrence is then

$$H^{l-1} = W_l^T D^l H^l D^l W_l.$$

Setting $l = L$ and using $H^L = 0$ yields $H^{L-1} = 0$. Re-applying the same identity inductively gives $H^{L-2} = 0, \ldots, H^1 = 0$; thus every activation-space block of the exact Hessian is zero.

The weight-space block for layer $l$ factorises as

$$H_{W_l} = (x^{l-1} x^{l-1T}) \otimes (D^l H^l D^l).$$

Since $D^l H^l D^l = 0$ once the upstream $H^l$ has been shown to vanish, each $H_{W_l}$ is also identically zero. Hence, away from the measure-zero kink set of the ReLU, the entire Hessian of the network collapses whenever the trace of the output-layer Hessian vanishes at a local optimiser.

**Lemma 11.** *[Vanishing of polynomial sharpness scores] Let $W = (W_{clf}, W_{repr})$ be the parameters of a feedforward relu network whose logits are $z = W_{clf}^T \phi_{W_{repr}}(x)$ and whose loss is soft-max cross-entropy. For any scale factor $\alpha > 1$ define the* logit–scaling *re-parameterisation*

$$W(\alpha) = (\alpha W_{clf}, W_{repr}).$$

*Assume the sharpness score $\mathcal{S} : (W, H(W)) \mapsto \mathbb{R}_{\geq 0}$ satisfies the polynomial growth bound*

$$\mathcal{S}(W, H) \leq C\,(1 + \|W\|_F)^r\,(1 + \|H\|_F)^s, \tag{6}$$

*for constants $C > 0$, $r > 0$, $s \geq 0$, where $H(W) = \nabla_W^2 \ell$ is the exact Hessian. Then*

$$\lim_{\alpha \to \infty} \mathcal{S}\big(W(\alpha), H(W(\alpha))\big) = 0.$$

*Proof.* Fix one training example and let $k$ be its correct class. Write the *margins* $\Delta_j = z_k - z_j$ for $j \neq k$ and $\Delta_{\min} = \min_{j \neq k} \Delta_j > 0$. After scaling the logits to $\alpha z$ the soft-max probabilities are $p_j(\alpha) = \mathrm{softmax}_j(\alpha z)$. For every $j \neq k$

$$p_j(\alpha) = \frac{e^{\alpha z_j}}{\sum_i e^{\alpha z_i}} \leq e^{-\alpha \Delta_j} \leq e^{-\alpha \Delta_{\min}}. \tag{1}$$

The logit-space Hessian for cross-entropy is $H_{ce} = diag(p) - pp^T$, so its trace is

$$\operatorname{tr} H_{ce}(\alpha) = p_k(1 - p_k) + \sum_{j \neq k} p_j(1 - p_j).$$

Using $1 - p_k = \sum_{j \neq k} p_j$ and $0 \leq 1 - p_j \leq 1$ we get

$$\operatorname{tr} H_{ce}(\alpha) \leq \sum_{j \neq k} p_j + \sum_{j \neq k} p_j = 2 \sum_{j \neq k} p_j(\alpha) \leq 2(K-1) e^{-\alpha \Delta_{\min}} =: D e^{-\alpha \Delta_{\min}}. \quad (2)$$

Since $\|H_{ce}\|_F \leq \sqrt{K} \operatorname{tr} H_{ce}$,

$$\|H_{ce}(\alpha)\|_F \leq D' e^{-\alpha \Delta_{\min}} \quad \text{for } D' = D\sqrt{K}. \quad (3)$$

Given the derivations above in Appendix B, that is the whole hessian is a linear function of the penultimate layer, there exists a $D'' > D'$ such that

$$\|H(W(\alpha))\|_F \leq D'' e^{-\alpha \Delta_{\min}} \quad (4)$$

Only the classifier block is scaled, hence

$$\|W(\alpha)\|_F = \sqrt{\alpha^2 \|W_{clf}\|_F^2 + \|W_{repr}\|_F^2} = \Theta(\alpha). \quad (5)$$

Insert (4) and (5) into equation 6:

$$\mathcal{S}\big(W(\alpha), H(\alpha)\big) \leq C \left(1 + A\alpha\right)^r \left(1 + D'' e^{-\alpha \Delta_{\min}}\right)^s \leq C' \alpha^r e^{-s\alpha \Delta_{\min}},$$

for constants $A, C'$. The exponential decay dominates the polynomial growth, implying $\mathcal{S}\big(W(\alpha), H(\alpha)\big) \to 0$ as $\alpha \to \infty$. $\qquad \square$

**Lemma 12.** *[Confidence–collapse for losses with at most quadratic poles] Let $x \in \mathbb{R}^d$ be the input to the classifier, $W \in \mathbb{R}^{K \times d}$ the classifier, $z = Wx$ the logits, and $p = \operatorname{softmax}(z)$ the class probabilities. Fix a twice–differentiable loss $\ell(p, y)$ and write*

$$h_i(p) = \frac{\partial^2 \ell}{\partial p_i^2}(p, y), \qquad i = 1, \dots, K.$$

*Assume there is $M > 0$ such that for every $p$ in the open simplex $\Delta^{K-1} := \{ p \in (0,1)^K \mid \sum_i p_i = 1\}$*

$$h_y(p) p_y^2 \leq M, \quad \max_{j \neq y} |h_j(p)| \leq M \quad (A)$$

*Denote $H_W = \nabla_W^2 \ell\big(p(Wx), y\big)$. Then, along any sequence of* finite *weight matrices for which the model becomes confident $(p_y \to 1^-)$,*

$$\|H_W\|_F \longrightarrow 0.$$

**Remark.** *The boundary point $p = e_y$ (true one-hot) is* not *reached with finite weights; it would require $z_y - z_j \to \infty$. The lemma therefore treats the limit $p_y \nearrow 1$ inside the open simplex, where all derivatives are well defined.*

*Proof.* Let $z = Wx \in \mathbb{R}^K$ and $p = \operatorname{softmax}(z) \in \Delta^{K-1}$. The Jacobian of the softmax is

$$J = \nabla_z p = \operatorname{diag}(p) - pp^\mathsf{T} \in \mathbb{R}^{K \times K}.$$

Write $r_i^\mathsf{T}$ for the $i$th row of $J$, so $r_i = p_i(e_i - p)$. Then $\|r_i\|_2^2 = (p_i^2(1 * 2p_i + \|p\|_2^2)) \leq p_i^2$. Let $g_i = \partial \ell / \partial p_i$ and $h_i = \partial^2 \ell / \partial p_i^2$. The full second-order chain rule gives

$$H_z := \nabla_z^2 \ell(p(z), y) = J^\mathsf{T} \operatorname{diag}(h) J + \sum_{i=1}^K g_i \nabla_z^2 p_i =: H_z^{(1)} + H_z^{(2)}.$$

We first bound $\|H_z^{(1)}\|_F$. Since $\mathrm{diag}(h)$ is diagonal, we have

$$H_z^{(1)} = \sum_{i=1}^{K} h_i r_i r_i^\mathsf{T}, \quad \text{so} \quad \|r_i r_i^\mathsf{T}\|_F = \|r_i\|_2^2.$$

Therefore,

$$\|H_z^{(1)}\|_F \leq \sum_{i \neq y} |h_i| \|r_i\|_2^2 + |h_y| \|r_y\|_2^2.$$

For $i \neq y$, assumption (A) gives $|h_i| \leq M$, so $|h_i| \|r_i\|_2^2 \leq M p_i^2$. For $i = y$, we compute

$$\|r_y\|_2^2 = p_y^2 \|e_y - p\|_2^2 = p_y^2 \left( (1 - p_y)^2 + \sum_{j \neq y} p_j^2 \right).$$

By assumption (A), $|h_y| \leq M/p_y^2$, we have

$$|h_y| \|r_y\|_2^2 \leq \frac{M}{p_y^2} \cdot p_y^2 \left( (1 - p_y)^2 + \sum_{j \neq y} p_j^2 \right) = M \left( (1 - p_y)^2 + \sum_{j \neq y} p_j^2 \right).$$

Since $p_j^2 \leq p_j$ and $\sum_{j \neq y} p_j = 1 - p_y$, it follows that

$$\sum_{j \neq y} p_j^2 \leq 1 - p_y,$$

so we conclude

$$|h_y| \|r_y\|_2^2 \leq M(1 - p_y)^2 + M(1 - p_y).$$

Since $p_j^2 \leq p_j$ and $\sum_{j \neq y} p_j = 1 - p_y$, we conclude

$$\|H_z^{(1)}\|_F \leq M \sum_{j \neq y} p_j^2 + M(1 - p_y) \leq 2M(1 - p_y).$$

Next, we bound $\|H_z^{(2)}\|_F$. The second derivative of softmax satisfies $|\nabla_z^2 p_i| \leq C_1 p_i$ entrywise for some fixed constant $C_1(K)$, hence

$$\|\nabla_z^2 p_i\|_F \leq C_2 p_i.$$

Therefore,

$$\|H_z^{(2)}\|_F \leq \sum_{i=1}^{K} |g_i| \|\nabla_z^2 p_i\|_F \leq C_2 \sum_{i=1}^{K} |g_i| p_i.$$

For all standard losses (cross-entropy, focal, squared-error, etc.), the product $|g_i| p_i$ is bounded as $p_y \to 1^-$. Thus there exists $G > 0$ such that

$$\|H_z^{(2)}\|_F \leq G(1 - p_y).$$

Combining both bounds, we have

$$\|H_z\|_F \leq \|H_z^{(1)}\|_F + \|H_z^{(2)}\|_F \leq (2M + G)(1 - p_y) \to 0 \quad \text{as } p_y \to 1^-.$$

Finally, since $z = Wx$, the Hessian with respect to the weights is $H_W = (xx^\mathsf{T}) \otimes H_z$, and

$$\|H_W\|_F = \|xx^\mathsf{T}\|_F \cdot \|H_z\|_F = \|x\|_2^2 \cdot \|H_z\|_F \to 0.$$

$\square$

## B.1 MORE CONTEXT TO LEMMA 12

Assumption (A) is easiest to interpret in terms of how fast the loss can blow up when a probability goes to 0. It allows at most a $\frac{1}{p_i^2}$ singularity–and only for the *true* class $y$; all other curvatures must remain bounded. That restriction may look arbitrary, yet the entries in Table 1 show it is satisfied by almost every loss routinely used for hard-label classification: negative log-likelihood, focal loss with $\gamma \geq 1$, the Brier score, and the (hard-label) KL divergence. For these objectives a perfectly confident model ($p_y \to 1$) is a well-behaved stationary point: the weight-space Hessian collapses to zero as stated in Lemma 12.

The only common exceptions are losses that deliberately assign positive target mass to *more than one* class. Label smoothing and KL to a soft target inject a factor $q_i/p_i^2$ into the second derivative for every class with $q_i > 0$; the same problem appears in reverse KL, $D_{\mathrm{KL}}(p\|q) = \sum_i p_i \log(p_i/q_i)$, whose curvature is $h_i = 1/p_i$. Because such terms diverge when $p_i \to 0$, Assumption (A) fails and Lemma 12 no longer applies i.e. the proof is not valid anymore. In this particular case; this is not a problem since $e_y$ is anyways not a local minima and hence not of interest to make statements about robustness and generalization. Yet this is an interesting result which could explain the training dynamics of teach-student optimisation approaches.

Hinge-style or margin losses defined on the logits rather than on the probabilities fall outside the scope of the lemma because they are either not twice differentiable or do not depend on $p$ at all.

| Loss function | $h_y(p)$ | $h_{j\neq y}(p)$ | Assumption (A) |
|---|---|---|---|
| Cross-entropy $\ell = -\log p_y$ | $\frac{1}{p_y^2}$ | 0 | ✓ |
| Focal $(1-p_y)^\gamma \log p_y$ | $\mathcal{O}(p_y^{-2+\gamma})$ | 0 | ✓ if $\gamma \geq 1$ |
| Brier / MSE $\sum_i (p_i - \delta_{iy})^2$ | 2 | 2 | ✓ |
| KL to hard label $\ell = D_{\mathrm{KL}}(\delta_y \parallel p)$ | $\frac{1}{p_y^2}$ | 0 | ✓ |
| KL to soft target $\ell = D_{\mathrm{KL}}(q \parallel p)$ | $\frac{q_y}{p_y^2}$ | $\frac{q_j}{p_j^2}$ | ✗ (some $q_j > 0$) |
| Reverse KL to soft target $\ell = D_{\mathrm{KL}}(p \parallel q)$ | $\frac{1}{p_y}$ | 0 | ✓ |

Table 1: Curvature terms $h_i(p)$ for common loss functions. Assumption (A) is violated when any $h_i(p)$ becomes unbounded on the simplex. We give the derivations in Section B.2.

## B.2 ADDITIONAL DERIVATIVES FOR TABLE 1

**Exact curvature expressions.** Let $h_i(p) = \frac{\partial^2 \ell}{\partial p_i^2}(p, y)$ be the second derivative of the loss w.r.t. class $i$ in the probability simplex.

**Focal loss.**

$$\ell_{\mathrm{focal}}(p, y; \gamma) = -(1-p_y)^\gamma \log p_y.$$

Only $p_y$ is involved, so $h_j = 0$ for $j \neq y$. Let $v = 1 - p_y$:

$$h_y(p) = v^{\gamma-2}\left[-\gamma(\gamma-1)\log p_y + \frac{2\gamma v}{p_y} + \frac{v^2}{p_y^2}\right].$$

Then,

$$p_y^2 h_y(p) = v^{\gamma-2}\left[-\gamma(\gamma-1)p_y^2 \log p_y + 2\gamma p_y v + v^2\right].$$

This remains bounded as $p_y \to 1$ if and only if $\gamma \geq 1$.

**KL divergence to soft target.**

$$\ell_{\mathrm{KL}}(p, q) = \sum_i q_i \log \frac{q_i}{p_i} = -\sum_i q_i \log p_i + \mathrm{const},$$

so

$$h_i(p) = \frac{q_i}{p_i^2}.$$

If any $q_j > 0$, then $h_j(p)$ diverges as $p_j \to 0$, and assumption (A) is violated.

## C    PROOFS OF THEORETICAL RESULTS

In this section, we provide the proofs for the theoretical results in this paper.

### C.1    PROOF OF LEMMA 7

For convenience, we restate the lemma.

**Lemma 7.** *Let $f = g(\mathbf{w}\phi(x))$ be a model with $\phi$ $L$-Lipschitz and $\|\phi(x)\| \geq r$, and $\xi, x \in \mathcal{X}$ with $\|\xi - x\| \leq \delta$, then there exists a $\Delta > 0$ with $\Delta \leq L\delta r^{-1}$, such that $\phi(\xi) = \phi(x) + \Delta A\phi(x)$, where $A$ is an orthogonal matrix.*

*Proof.* It follows from the proof in Thm. 5 in Petzka et al. (2021) that we can represent any vector $v \in \mathcal{X}$ as $v = w + \Delta Aw$ for some vector $w \in X$, $\Delta \in \mathbb{R}_+$ and $A$ an orthogonal matrix. Then

$$\phi(\xi) = \phi(x) + \Delta A\phi(x)$$
$$\Leftrightarrow (\phi(\xi) - \phi(x)) = \Delta(A\phi(x))$$
$$\Leftrightarrow \Delta \leq \|(\phi(\xi) - \phi(x))\|\|(A\phi(x))^{-1}\| = \underbrace{\|(\phi(x + \Delta'A'x) - \phi(x))\|}_{\leq L\Delta'}\|(A\phi(x))^{-1}\|$$

$$\Leftrightarrow \Delta \leq L\Delta'\|(A\phi(x))^{-1}\| \underbrace{\leq}_{A \text{ orth.}} L\Delta'\frac{1}{r} .$$

The result follows from $\|\xi - x\| = \Delta' \leq \delta$. $\qquad\qquad\qquad\qquad\qquad\qquad\qquad\qquad\square$

### C.2    PROOF OF PROPOSITION 8

For convenience, we restate the proposition.

**Proposition 8.** *For $(x, y) \in \mathcal{X} \times \mathcal{Y}$ with $\|x\| \leq 1$ for all $x \in \mathcal{X}$, a model $f(x) = g(\mathbf{w}\phi(x))$ at a minimum $\mathbf{w} \in \mathbb{R}^{m \times k}$ with $\phi$ $L$-Lipschitz and $\|\phi(x)\| \geq r$, and the cross-entropy loss $\ell(\mathbf{w}) = \ell(g(\mathbf{w}\phi(x)), y)$ of $f$ on $(x, y)$, it holds for all $\xi \in \mathcal{X}$ with $\|x - \xi\|_2 \leq \delta$ that*

$$\ell(f(\xi), y) - \ell(f(x), y) \leq \frac{\delta^2}{2r^2}L^2\kappa_{Tr}^\phi(\mathbf{w}) + \frac{\delta^3}{24r^3}kmL^6 .$$

*Proof.* The remainder $R_2$ in Eq. 5 is

$$R_2(\mathbf{w}, \Delta) \leq \sup_{\substack{h \in \mathbb{R}^m \\ \|h\|=1}} \sup_{c \in (0,1)} \frac{\Delta^3}{3!} \sum_{i,j,k}^d \frac{\partial^3\ell}{\partial w_i\partial w_j\partial w_k}(x + c\Delta h) ,$$

where $w \in \mathbb{R}^d$ with $d = km$ is the vectorization of $\mathbf{w} \in \mathbb{R}^{m \times k}$. We now bound this remainder. Using the representation of the loss Hessian from Eq. 4, we can write the partial third derivatives in the remainder as

$$\frac{\partial^3\ell}{\partial w_i\partial w_j\partial w_k}(x) = \sum_{\substack{o,l,j \in [k] \\ a,b,c \in [m]}} -\left(\hat{y}_j\hat{y}_o(\mathbb{1}_{o=j} - \hat{y}_l) + \hat{y}_l\hat{y}_o(\mathbb{1}_{o=l} - \hat{y}_j)\right)\phi(x)_a\phi(x)_b\phi(x)_c ,$$

where $\hat{y} = f(x)$. Under the assumption that $\phi$ is $L$-Lipschitz, for all $x \in \mathcal{X}$, $\|x\| \leq 1$ and observing that $\sum_{o \in [k]} \hat{y}_o = 1$ we can bound this term by

$$\frac{\partial^3\ell}{\partial w_i\partial w_j\partial w_k}(x) \leq \frac{1}{4}kmL'^3 . \qquad\qquad\qquad\qquad (7)$$

The terms $k, m$ follow from the sum over all rows and columns of $\mathbf{w}$, and the factor $4^{-1}$ follows from the fact that the predictions in $\hat{y}$ sum up to $1$. The factor $L'^3$ can be derived as follows.

$$\phi(x)_i \leq ||\phi(x)_i|| = ||(\phi(x)_i - \phi(\mathbf{0})_i) + \phi(\mathbf{0})_i|| \tag{8}$$

$$\leq ||(\phi(x)_i - \phi(\mathbf{0})_i)|| + ||\phi(\mathbf{0})_i|| \tag{9}$$

$$\leq L||x - \mathbf{0}|| + ||\phi(\mathbf{0})_i|| \qquad (\phi \text{ is } L\text{-Lipschitz}) \tag{10}$$

$$\leq L + ||\phi(\mathbf{0})_i|| \leq L + C_{\phi(\mathbf{0})} \qquad (||x|| \leq 1) \tag{11}$$

$$\tag{12}$$

where $C_{\phi(\mathbf{0})} := max_i||\phi(\mathbf{0})_i||$. For Relu networks without bias $C_{\phi(\mathbf{0})}$ is $0$ and empirically for networks with a bias term, it is very small i.e. $\approx 1$. Therefore $L' = L + C_{\phi(\mathbf{0})} \approx L$, in particular since for most realistic neural networks $L$ is large. For simplicity, we subsitute $L = L'$.

Inserting this bound into Eq. 5 yields

$$|\ell(f(\xi), y) - \ell(f(x), y)| \leq \Delta||\mathbf{w}||_F ||\nabla_{\mathbf{w}}\ell(\mathbf{w})||_F + \frac{\Delta^2}{2}\kappa^\phi_{Tr}(\mathbf{w}) + \frac{\Delta^3}{3!}\frac{1}{4}kmL^3 \ .$$

The result follows from setting $\Delta \leq L\delta r^{-1}$. □

## C.3 PROOF OF PROPOSITION 9

**Proposition 12.** *For a dataset $S \subset \mathcal{X} \times \mathcal{Y}$ with $\|x\| \leq 1$ for all $x \in \mathcal{X}$, a model $f(x) = g(\mathbf{w}\phi(x))$ at a minimum $\mathbf{w} \in \mathbb{R}^{m \times k}$ wrt. S, with $\phi$ L-Lipschitz and $\|\phi(x)\| \geq r$, and a loss function $\ell(\mathbf{w}) = \ell(g(\mathbf{w}\phi(x)), y)$ of $f$ on $(x, y)$, $d$ being the L2-distance, and $\epsilon > 0$, $f$ is $(\epsilon, \delta, S)$-robust against adversarial examples with*

$$\delta \geq \left( -\frac{8r^3k^3m^3L^9 + 27\epsilon}{27L^3\kappa^\phi_{Tr}(\mathbf{w})} + \left( -\frac{2^7}{27}\frac{r^6k^6m^6L^3}{\kappa^\phi_{Tr}(\mathbf{w})^6} + \frac{r^6\epsilon^2}{L^6\kappa^\phi_{Tr}(\mathbf{w})^2} - \frac{2^4r^6\epsilon k^3m^3L^{\frac{3}{2}}}{27\kappa^\phi_{Tr}(\mathbf{w})^4} \right)^{\frac{1}{2}} \right)^{\frac{1}{3}}$$

$$+ \left( -\frac{8r^3k^3m^3L^9 + 27\epsilon}{27L^3\kappa^\phi_{Tr}(\mathbf{w})} - \left( -\frac{2^7}{27}\frac{r^6k^6m^6L^3}{\kappa^\phi_{Tr}(\mathbf{w})^6} + \frac{r^6\epsilon^2}{L^6\kappa^\phi_{Tr}(\mathbf{w})^2} - \frac{2^4r^6\epsilon k^3m^3L^{\frac{3}{2}}}{27\kappa^\phi_{Tr}(\mathbf{w})^4} \right)^{\frac{1}{2}} \right)^{\frac{1}{3}}$$

$$+ \frac{2rkmL^2}{72\kappa^\phi_{Tr}(\mathbf{w})}$$

*where $\kappa_{Tr}(\mathbf{w})$ is the relative flatness of $f$ wrt. $\mathbf{w}$. That is,*

$$\delta \propto \frac{\epsilon^{\frac{1}{3}}}{(L^3\kappa_{Tr}(\mathbf{w}))^{\frac{1}{3}}} + \frac{rkmL^2}{\kappa_{Tr}(\mathbf{w})} \ .$$

*Proof.* From Prop. 8 it follows that we achieve $(\epsilon, \Delta, S)$-robustness where

$$\epsilon = \frac{\Delta^2}{2}\kappa^\phi_{Tr}(\mathbf{w}) + \frac{\Delta^3}{24}kmL^3 \ .$$

First, we need to solve this cubic equation for $\Delta$. For that, we substitute $a = \frac{kmL^3}{24}$ and $b = \frac{\kappa^\phi_{Tr}(w)}{2}$ and get

$$0 = a\Delta^3 + b\Delta^2 - \epsilon = \Delta^3 + \frac{a}{b}\Delta^2 - \frac{\epsilon}{b} = \Delta^3 + \alpha\Delta^2 - \beta \ ,$$

by subsituting $\alpha = \frac{a}{b}$ and $\beta = \frac{\epsilon}{b}$. We use the depressed cubic form $\Delta = t - \frac{\alpha}{3}$ and get

$$0 = \left( t - \frac{\alpha}{3} \right)^3 + \alpha \left( t - \frac{\alpha}{3} \right)^2 - \beta$$

$$\Longleftarrow 0 = t^3 - t^2\alpha + t\frac{\alpha^2}{3} - \frac{\alpha^3}{27} + t^2\alpha - \frac{2\alpha^2 t}{3} + \frac{\alpha^3}{9} - \beta$$

$$\Longleftarrow 0 = t^3 - t\frac{\alpha^2}{3} + \frac{2\alpha^3}{27} - \beta \ .$$

with $p = -\frac{\alpha^2}{3}$ and $q = \frac{2\alpha^3}{27} - \beta$ we get the form $0 = t^3 + pt + q$ for which we can apply Cardano's formula.

$$t = \left(-\frac{q}{2} + \left(\frac{q^2}{4} + \frac{p^3}{9}\right)^{\frac{1}{2}}\right)^{\frac{1}{3}} + \left(-\frac{q}{2} - \left(\frac{q^2}{4} + \frac{p^3}{9}\right)^{\frac{1}{2}}\right)^{\frac{1}{3}}$$

Resubstituting $p, q, \alpha, \beta$ yields

$$\frac{q}{2} = \frac{a^3}{27b^3} - \frac{\epsilon}{2n} \quad , \quad \frac{p^3}{9} = -\frac{1}{9^3}\frac{a^6}{b^6} \quad , \quad \frac{q^2}{4} = \frac{1}{27^2}\frac{a^6}{b^6} + \frac{\epsilon^2}{4b^2} - \frac{\alpha^3\epsilon}{27b^4} \quad .$$

Substituting this in the solution for $t$ then gives

$$t = \left(-\frac{a^3}{27b^3} + \frac{\epsilon}{2b} + \left(-\frac{2a^6}{27b^6} + \frac{\epsilon^2}{4b^2} - \frac{a^3\epsilon}{27b^4}\right)^{\frac{1}{2}}\right)^{\frac{1}{3}}$$

$$+ \left(-\frac{a^3}{27b^3} + \frac{\epsilon}{2b} - \left(-\frac{2a^6}{27b^6} + \frac{\epsilon^2}{4b^2} - \frac{a^3\epsilon}{27b^4}\right)^{\frac{1}{2}}\right)^{\frac{1}{3}}$$

Substituting $a, b$ and $t = \Delta + \frac{\alpha}{3}$ yields

$$\Delta = \left(-\frac{8k^3m^3L^9 + 27\epsilon}{27\kappa_{Tr}^\phi(\mathbf{w})} + \left(-\frac{2^7}{27}\frac{k^6m^6L^{18}}{\kappa_{Tr}^\phi(\mathbf{w})^6} + \frac{\epsilon^2}{\kappa_{Tr}^\phi(\mathbf{w})^2} - \frac{2^4\epsilon k^3m^3L^9}{27\kappa_{Tr}^\phi(\mathbf{w})^4}\right)^{\frac{1}{2}}\right)^{\frac{1}{3}}$$

$$+ \left(-\frac{8k^3m^3L^9 + 27\epsilon}{27\kappa_{Tr}^\phi(\mathbf{w})} - \left(-\frac{2^7}{27}\frac{k^6m^6L^{18}}{\kappa_{Tr}^\phi(\mathbf{w})^6} + \frac{\epsilon^2}{\kappa_{Tr}^\phi(\mathbf{w})^2} - \frac{2^4\epsilon k^3m^3L^9}{27\kappa_{Tr}^\phi(\mathbf{w})^4}\right)^{\frac{1}{2}}\right)^{\frac{1}{3}}$$

$$+ \frac{2kmL^3}{72\kappa_{Tr}^\phi(\mathbf{w})}$$

Finally, from Lemma 7 we have $\Delta \leq L\delta r^{-1}$, so that

$$\delta \geq \frac{r}{L}\left(-\frac{8k^3m^3L^9 + 27\epsilon}{27\kappa_{Tr}^\phi(\mathbf{w})} + \left(-\frac{2^7}{27}\frac{k^6m^6L^{18}}{\kappa_{Tr}^\phi(\mathbf{w})^6} + \frac{\epsilon^2}{\kappa_{Tr}^\phi(\mathbf{w})^2} - \frac{2^4\epsilon k^3m^3L^9}{27\kappa_{Tr}^\phi(\mathbf{w})^4}\right)^{\frac{1}{2}}\right)^{\frac{1}{3}}$$

$$+ \frac{r}{L}\left(-\frac{8k^3m^3L^9 + 27\epsilon}{27\kappa_{Tr}^\phi(\mathbf{w})} - \left(-\frac{2^7}{27}\frac{k^6m^6L^{18}}{\kappa_{Tr}^\phi(\mathbf{w})^6} + \frac{\epsilon^2}{\kappa_{Tr}^\phi(\mathbf{w})^2} - \frac{2^4\epsilon k^3m^3L^9}{27\kappa_{Tr}^\phi(\mathbf{w})^4}\right)^{\frac{1}{2}}\right)^{\frac{1}{3}}$$

$$+ \frac{2rkmL^2}{72\kappa_{Tr}^\phi(\mathbf{w})}$$

$$= \left(-\frac{8r^3k^3m^3L^9 + 27\epsilon}{27L^3\kappa_{Tr}^\phi(\mathbf{w})} + \left(-\frac{2^7}{27}\frac{r^6k^6m^6L^3}{\kappa_{Tr}^\phi(\mathbf{w})^6} + \frac{r^6\epsilon^2}{L^6\kappa_{Tr}^\phi(\mathbf{w})^2} - \frac{2^4r^6\epsilon k^3m^3L^{\frac{3}{2}}}{27\kappa_{Tr}^\phi(\mathbf{w})^4}\right)^{\frac{1}{2}}\right)^{\frac{1}{3}}$$

$$+ \left(-\frac{8r^3k^3m^3L^9 + 27\epsilon}{27L^3\kappa_{Tr}^\phi(\mathbf{w})} - \left(-\frac{2^7}{27}\frac{r^6k^6m^6L^3}{\kappa_{Tr}^\phi(\mathbf{w})^6} + \frac{r^6\epsilon^2}{L^6\kappa_{Tr}^\phi(\mathbf{w})^2} - \frac{2^4r^6\epsilon k^3m^3L^{\frac{3}{2}}}{27\kappa_{Tr}^\phi(\mathbf{w})^4}\right)^{\frac{1}{2}}\right)^{\frac{1}{3}}$$

$$+ \frac{2rkmL^2}{72\kappa_{Tr}^\phi(\mathbf{w})}$$

$\square$

## C.4 Derivation of Hessian and Third Derivative

Let $\phi \in \mathbb{R}^m$ denote the embedding of the feature extractor and $W \in \mathbb{R}^{K \times m}$, where we denote the weights of the $k$-th neuron as $w_k$. The output layer is given by the softmax function $\hat{y}_k = \text{softmax}(W\phi) \in \mathbb{R}^K$. More precisely the softmax is given by,

$$\hat{y}_k = \frac{\exp(w_k\phi)}{\sum_{j=1}^{K}\exp(w_j\phi)}$$

For simplicity, we omit the bias term. The one-hot encoded ground truth is given by $y$. The derivative of the loss $L$ function wrt. the weight vector $w_j$ can be computed as

$$\frac{\partial L(y,\hat{y})}{\partial w_j} = -(y_j - \hat{y}_j)\phi^T$$

**Second derivative**

$$\begin{aligned}
\frac{\partial L(y,\hat{y})}{\partial w_l\,\partial w_j} &= \frac{\partial}{\partial w_l} - (y_j - \hat{y}_j)\phi^T \\
&= \frac{\partial}{\partial w_l} - y_j\phi^T + \frac{\partial}{\partial w_l}\hat{y}_j\phi^T \\
&= \frac{\partial}{\partial w_l}\hat{y}_j\phi^T
\end{aligned}$$

The last equation follows from $y$ being independent of $w_l$. Next, we do a case analysis on $l = j$.

- ( $l = j$): In (1), we use the quotient rule and in (2) definition of softmax.

$$\begin{aligned}
\frac{\partial}{\partial w_l}\hat{y}_j\phi^T &= \frac{\partial}{\partial w_j}\frac{\exp(w_j\phi)}{\sum_{k=1}^{K}\exp(w_k\phi)}\phi^T \\
&= \frac{(\exp(w_j\phi)\sum_{k=1}^{K}\exp(w_k\phi))\phi\phi^T - \exp(w_j\phi)\exp(w_j\phi)\phi\phi^T}{(\sum_{k=1}^{K}\exp(w_k\phi))^2} \quad (1) \\
&= \left(\frac{\exp(w_j\phi)\sum_{k=1}^{K}\exp(w_k\phi)}{(\sum_{k=1}^{K}\exp(w_k\phi))^2} - \frac{\exp(w_j\phi)^2}{(\sum_{k=1}^{K}\exp(w_k\phi))^2}\right)\phi\phi^T \\
&= (\hat{y}_j - \hat{y}_j^2)\phi\phi^T \in \mathbb{R}^{m\times m} \quad (2)
\end{aligned}$$

- ( $l \neq j$): Again quotient rule, but the left side vanishes.

$$\frac{\partial}{\partial w_l}\hat{y}_j\phi^T = -\hat{y}_l\hat{y}_j\phi\phi^T \in \mathbb{R}^{m\times m}$$

Then we have

$$\frac{\partial L(y,\hat{y})}{\partial w_l\,\partial w_j} = \hat{y}_l(\mathbb{1}_{[l=j]} - \hat{y}_j)\phi\phi^T \in \mathbb{R}^{m\times m}$$

The hessian is then given by

$$H(L;W)(y,\hat{y}) = (\mathrm{diag}(\hat{y}) - \hat{y}\hat{y}^T) \otimes \phi\phi^T \in \mathbb{R}^{Km\times Km}$$

**Third derivative**  First, rewrite

$$\frac{\partial L(y,\hat{y})}{\partial w_l\,\partial w_j} = \hat{y}_l(\mathbb{1}_{[l=j]} - \hat{y}_j)\phi\phi^T = \hat{y}_l\mathbb{1}_{[l=j]}\phi\phi^T - \hat{y}_l\hat{y}_j\phi\phi^T$$

Then we define a new operator $\Pi : \mathbb{R}^n \times \mathbb{R}^m \times \mathbb{R}^o \to \mathbb{R}^{n\times m\times o}$, $\Pi(x,y,z)_{ijk} = x_i y_j z_k$. We now compute

$$\frac{\partial L(y,\hat{y})}{\partial w_o\,\partial w_l\,\partial w_j} = \frac{\partial}{\partial w_o}\hat{y}_l\mathbb{1}_{[l=j]}\phi\phi^T - \hat{y}_l\hat{y}_j\phi\phi^T$$

Again we make a CA on $j = l$

- ( $l = j$ ):

$$\frac{\partial}{\partial w_o}(\hat{y}_l\phi\phi^T - \hat{y}_l^2\phi\phi^T) = \frac{\partial}{\partial w_o}\hat{y}_l\phi\phi^T - \frac{\partial}{\partial w_o}\hat{y}_l^2\phi\phi^T$$
$$= \hat{y}_o(\mathbb{1}_{[o=l]} - \hat{y}_l)\,\Pi(\phi,\phi,\phi) - 2(\hat{y}_o(\mathbb{1}_{[o=l]} - \hat{y}_l)\,\Pi(\phi,\phi,\phi))$$
$$= -\hat{y}_o(\mathbb{1}_{[o=l]} - \hat{y}_l)\,\Pi(\phi,\phi,\phi)$$

- ( $l \neq j$ ):

$$\frac{\partial}{\partial w_o} - \hat{y}_l\hat{y}_j\phi\phi^T = -(\frac{\partial}{\partial w_o}\hat{y}_l)\hat{y}_j\phi\phi^T - \hat{y}_l(\frac{\partial}{\partial w_o}\hat{y}_j)\phi\phi^T$$
$$= -\hat{y}_j\hat{y}_o(\mathbb{1}_{[o=l]} - \hat{y}_l)\cdot\Pi(\phi,\phi,\phi) - \hat{y}_l\hat{y}_o(\mathbb{1}_{[o=i]} - \hat{y}_j)\cdot\Pi(\phi,\phi,\phi)$$
$$= -[\hat{y}_j\hat{y}_o(\mathbb{1}_{[o=l]} - \hat{y}_l) + \hat{y}_l\hat{y}_o(\mathbb{1}_{[o=i]} - \hat{y}_j)]\cdot\Pi(\phi,\phi,\phi) \in \mathbb{R}^{m\times m\times m}$$
$$\rightarrow -[\hat{y}_j\hat{y}_o(\mathbb{1}_{[o=l]} - \hat{y}_l) + \hat{y}_l\hat{y}_o(\mathbb{1}_{[o=j]} - \hat{y}_j)]_{j,l,o=1..k} \otimes \Pi(\phi,\phi,\phi) \in \mathbb{R}^{Km\times Km\times Km}$$

# D  ADDITIONAL RESULTS

## D.1  RESULTS FOR OTHER ARCHITECTURES AND DATASETS

We provide the remaining results in Figure 7 to Figure 12. We see the same trend as observed in the main paper. The only difference is that for models trained CIFAR-100 the basin width correlates not as strong with the sharpness at 0 compared to models trained CIFAR-10.

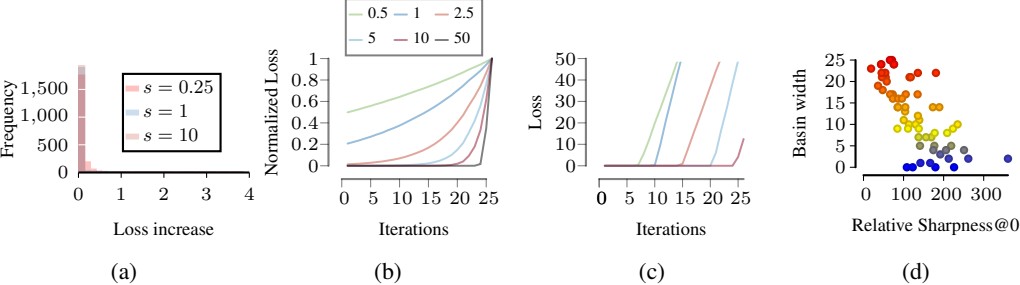

| (a) | (b) | (c) | (d) |

Figure 5: *Loss geometry for* VGG11.**(a)** We report the distribution of the loss increase for varying scaling values. **(b)** We show for one example how the basin formes as we increase the scaling. We use the normalized loss to plot all in one axis. **(c)** We show examples of samples exhibiting different basin widths. **(d)** We plot the sharpness measured at the test data ($s = 1$) and the basin width.

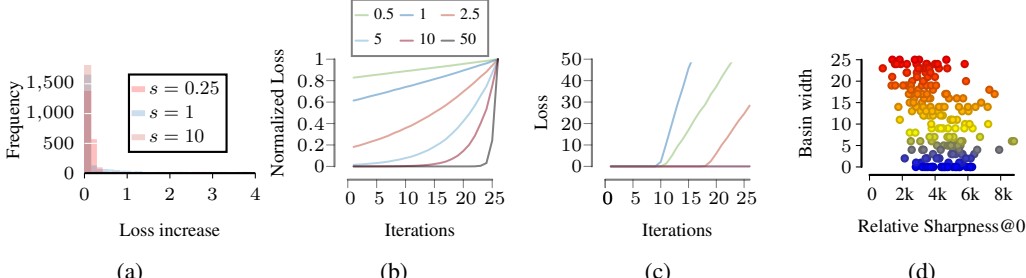

| (a) | (b) | (c) | (d) |

Figure 6: *Loss geometry for* VGG11 on CIFAR-100.**(a)** We report the distribution of the loss increase for varying scaling values. **(b)** We show for one example how the basin formes as we increase the scaling. We use the normalized loss to plot all in one axis. **(c)** We show examples of samples exhibiting different basin widths. **(d)** We plot the sharpness measured at the test data ($s = 1$) and the basin width.

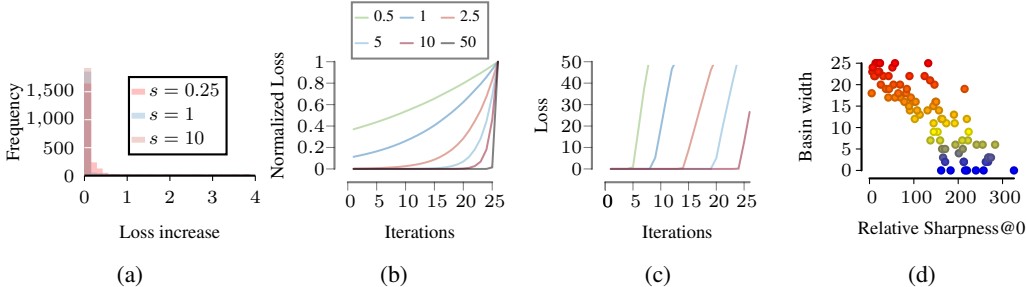

Figure 7: *Loss geometry for* RESNET-18. **(a)** We report the distribution of the loss increase for varying scaling values. **(b)** We show for one example how the basin formes as we increase the scaling. We use the normalized loss to plot all in one axis. **(c)** We show examples of samples exhibiting different basin widths. **(d)** We plot the sharpness measured at the test data ($s = 1$) and the basin width.

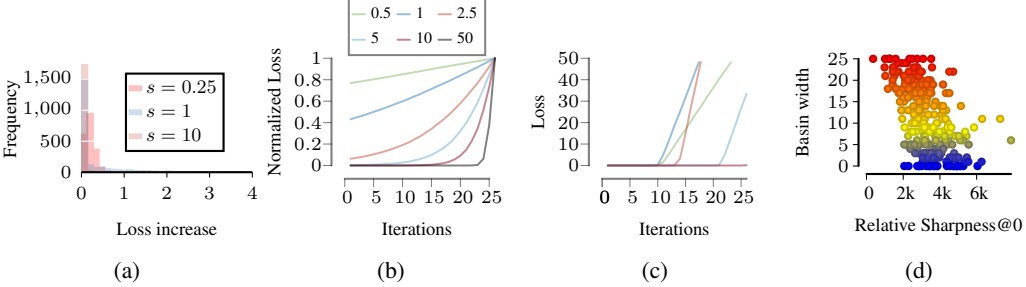

Figure 8: *Loss geometry for* RESNET-18 on CIFAR-100. **(a)** We report the distribution of the loss increase for varying scaling values. **(b)** We show for one example how the basin formes as we increase the scaling. We use the normalized loss to plot all in one axis. **(c)** We show examples of samples exhibiting different basin widths. **(d)** We plot the sharpness measured at the test data ($s = 1$) and the basin width.

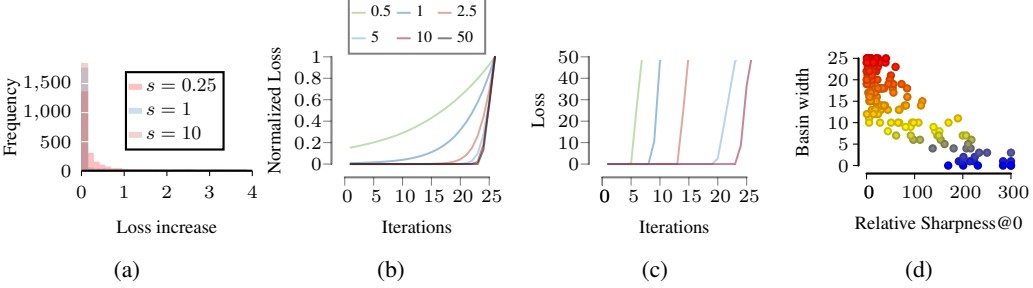

Figure 9: *Loss geometry for* WIDERESNET-28-4. **(a)** We report the distribution of the loss increase for varying scaling values. **(b)** We show for one example how the basin formes as we increase the scaling. We use the normalized loss to plot all in one axis. **(c)** We show examples of samples exhibiting different basin widths. **(d)** We plot the sharpness measured at the test data ($s = 1$) and the basin width.

## D.2 FLATNESS CAN INDUCE GRADIENT MASKING

To test whether the geometry backpropagates to the input, we need to compute the Hessian with respect to the input, which is prohibitively expensive. Instead, we take an actionable route: if the loss surface is truly flat, then first-order attacks must fail, as without curvature the optimization cannot succeed i.e. they become unattackable. We attack ResNet-18 models with standard PGD-$\ell_\infty$ (10 steps, $\epsilon = 8/255$, step size $\alpha = 2/255$) while varying the penultimate layer scaling factor $s$

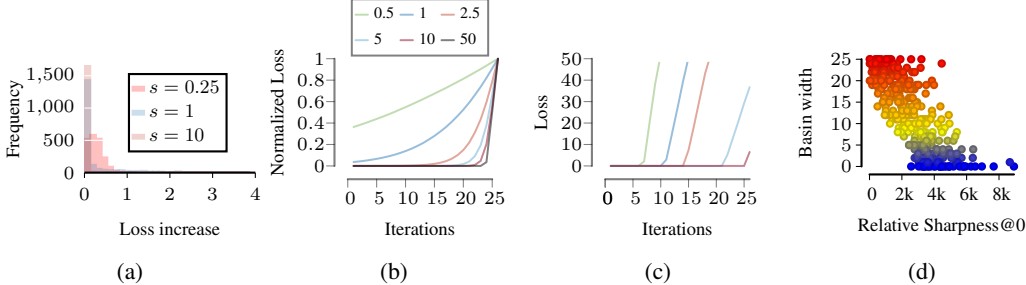

Figure 10: *Loss geometry for* WIDERESNET-28-4 *on CIFAR-100.***(a)** We report the distribution of the loss increase for varying scaling values. **(b)** We show for one example how the basin formes as we increase the scaling. We use the normalized loss to plot all in one axis. **(c)** We show examples of samples exhibiting different basin widths. **(d)** We plot the sharpness measured at the test data $(s = 1)$ and the basin width.

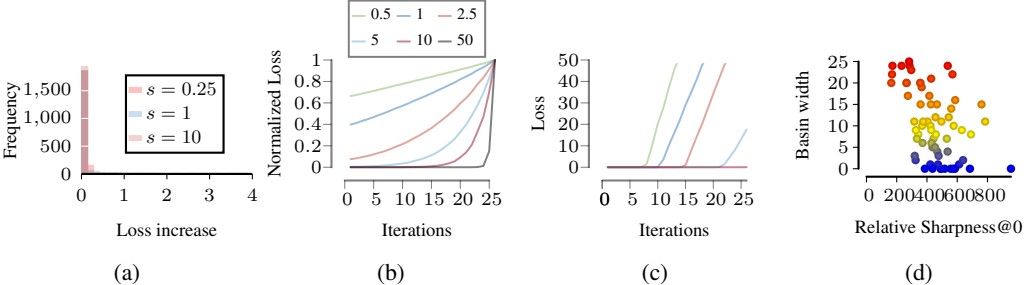

Figure 11: *Loss geometry for* DENSENET121.***(a)** We report the distribution of the loss increase for varying scaling values. **(b)** We show for one example how the basin formes as we increase the scaling. We use the normalized loss to plot all in one axis. **(c)** We show examples of samples exhibiting different basin widths. **(d)** We plot the sharpness measured at the test data $(s = 1)$ and the basin width.

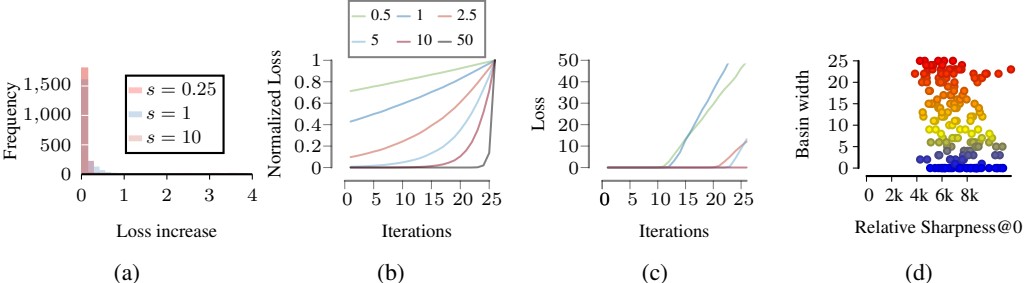

Figure 12: *Loss geometry for* DENSENET121 *on CIFAR-100.***(a)** We report the distribution of the loss increase for varying scaling values. **(b)** We show for one example how the basin formes as we increase the scaling. We use the normalized loss to plot all in one axis. **(c)** We show examples of samples exhibiting different basin widths. **(d)** We plot the sharpness measured at the test data $(s = 1)$ and the basin width.

from 1 to 100. We measure the robust test accuracy and average sharpness at the clean examples and show the results in Figure 13.

As we can see, increasing $s$ dramatically improves robust test accuracy. The standard ResNet with $s = 1$ achieves only 0% robust accuracy, while its scaled counterpart with $s = 100$ reaches 93%, which recovers the clean accuracy. Crucially, the relative sharpness measured at the penultimate layer strongly correlates with robust test accuracy across all scaling values. This direct correlation demonstrates that penultimate layer geometry indeed propagates to the input space. Importantly,

this does not mean that scaled networks are adversarially robust to according to Defintion 3Instead, the flat geometry prevents first-order attacks from finding adversarial examples. More specifically, adversarial examples found at $s = 1$ transfer with 100% success to other scales, indicating that the same vulnerabilities remain but are harder to reach. *This highlights that analyzing adversarial robustness through sharpness is more nuanced than it appears and requires careful evaluation.*

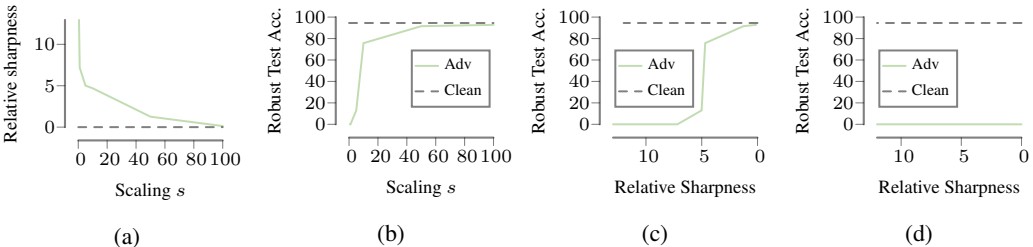

(a)         (b)         (c)         (d)

Figure 13: *Unattackable networks.* **(a)** we report the relative sharpness of the scaled networks at the test data. **(b)** Robust Test Accuracy with varying scaling $s$. **(c)** relative sharpness at the clean data vs. Robust Test Accuracy. **(d)** transferability of the attacks created on $s = 1$.

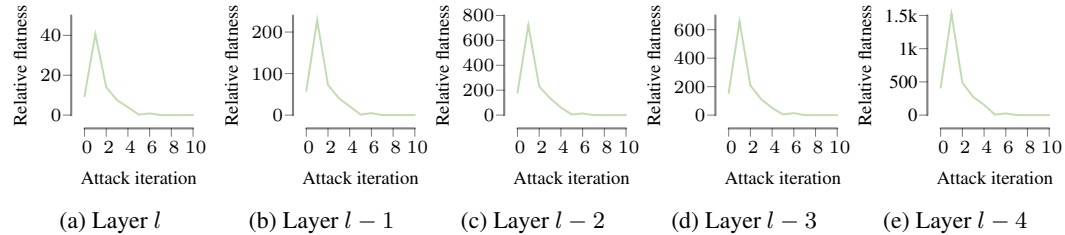

(a) Layer $l$   (b) Layer $l - 1$   (c) Layer $l - 2$   (d) Layer $l - 3$   (e) Layer $l - 4$

Figure 14: We show the relative sharpness measure computed in the penultimate layer $l$ and in shallower layers $l - 1$ to $l - 4$ for WIDERESNET-28-4. Due to memory and runtime constraints, we approximate the measure using Hutchinson trace estimation used in Petzka et al. (2021) on 500 images. We observe the same phenomena as in the penultimate layer, which justifies that we focus only on the penultimate layer for our theoretical and experimental analysis.

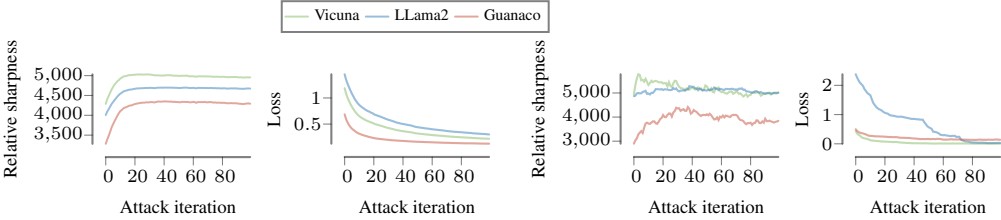

Figure 15: **Top**. We plot the relative sharpness and loss of the adversarial prompt for Viucna, LLama2 and Guanaco when attacked by the method of Zou et al. (2023). **Bottom**.We give per model example trajectories that first get sharper and then flatter again.

## D.3 DETECTING ADVERSARIAL EXAMPLES

It is possible to detect adversarial examples using a simple threshold on the relative sharpness measure. We did not include a practical study of this since it would go beyond the scope of this paper: Developing a sound method requires more than fine-tuning the threshold. Moreover, this requires comparing the approach to a wide range of state-of-the-art detection methods, which would be a paper of its own. Therefore, we leave this interesting practical aspect for future work. Nonetheless, we provide preliminary results. We trained a decision stump on the sharpness of clean and adversarial samples on CIFAR-10 for WIDERESNET-28-4 using 5-fold cross-validation, which yields the

following accuracies: $[0.92, 0.92, 0.93, 0.92, 0.92]$, i.e., adversarial examples can be detected with an average accuracy of $0.92$ with little to no difference between the folds.

## E    RELATED WORK

**Defenses and Guarantees**    The most widely used method for improving robustness to adversarial examples is adversarial training, which integrates adversarial perturbations into the training process (Szegedy et al., 2014; Shafahi et al., 2019; Kumari et al., 2019; Perolat et al., 2018; Shafahi et al., 2020; Cai et al., 2018; Tramèr et al., 2018; Wu et al., 2020; Carmon et al., 2019). While this approach enhances adversarial robustness, it frequently leads to a reduction in clean accuracy (Tsipras et al., 2018; Rade & Moosavi-Dezfooli, 2021; Nakkiran, 2019; Zhang et al., 2019). This trade-off remains controversial, prompting the question of whether it is fundamentally impossible to simultaneously achieve high performance and robustness.

One hypothesis is that adversarial training is inherently more difficult than standard training, and that networks simply underperform when faced with this more challenging task (Schmidt et al., 2018). Conversely, Raghunathan et al. (2020) argue that deep neural networks are capable of fitting even random noise (Zhang et al., 2017), suggesting that insufficient regularization, rather than task complexity, may be the root cause of this performance drop. Indeed, numerous works have explored strategies to improve generalization and thereby enhance the effectiveness of adversarial training. These include incorporating additional data (Carmon et al., 2019; Alayrac et al., 2019; Hendrycks et al., 2019; Gowal et al., 2021), using early stopping (Rice et al., 2020), and promoting flatness in the loss landscape (Wu et al., 2020; Stutz et al., 2021; Moosavi-Dezfooli et al., 2019; Xu et al., 2020).

To better understand this controversy and nature of adversarial vulnerability, a line of research has proposed the *in- and off-manifold hypothesis* (Stutz et al., 2019; Shamir et al., 2021; Zhang et al., 2022; Haldar et al., 2024; Melamed et al., 2023). According to this perspective, adversarial examples can be either on the data manifold or off it. Robust generalization is achievable primarily for in-manifold adversarial examples, while off-manifold ones tend to degrade clean accuracy during adversarial training. Song et al. (2018) support this view empirically, proposing a defense mechanism that learns the data manifold and projects adversarial examples onto it, thereby restoring correct classification.

This perspective requires additional assumptions about the data distribution. In practice, we often have access to only a finite and relatively small dataset—especially when compared to the high dimensionality of input spaces—which limits our ability to precisely model or leverage the underlying manifold structure.

From the other side, a natural approach to controlling model behavior on unseen samples is through the Lipschitz constant: if an input is close to a training sample, the model's prediction should not vary significantly. However, applying Lipschitz constraints directly to modern deep networks proves challenging. Global Lipschitz constants (Tsuzuku et al., 2018) tend to be too coarse to be meaningful, while local Lipschitz constants (Hein & Andriushchenko, 2017) are computationally intractable to estimate accurately. Moreover, Liang & Huang (2021) show that a large global Lipschitz constant does not necessarily imply large local constants, meaning that a model may still be locally robust despite a high global value. Conversely, enforcing a small global Lipschitz constant can help control local behavior but often leads to degraded clean accuracy. Yang et al. (2020) suggest that improving local Lipschitz constants alongside generalization could yield both robust and accurate models, though achieving this balance is not straightforward in practice.

Efforts to explicitly control Lipschitz properties during training have inspired the development of certification-based robustness techniques. In these methods, models are trained to produce consistent predictions under random input perturbations, which yields certified guarantees of robustness on training samples (Cohen et al., 2019; Salman et al., 2019). Interestingly, Kanai et al. (2023) analyze robustness in both the input and parameter spaces, concluding that enforcing smoothness in the input space often results in a non-flat loss surface with respect to parameters, which in turn harms generalization. This suggests that input-space robustness alone may not suffice to preserve clean accuracy, while parameter-space flatness alone may be insufficient for adversarial robustness. This tension ties into the broader understanding of generalization: robustness in parameter space—measured via loss

surface flatness—is known to correlate with improved generalization (Hochreiter & Schmidhuber, 1994; Liang et al., 2019; Tsuzuku et al., 2020; Foret et al., 2020; Petzka et al., 2021). Accordingly, enforcing smoothness in both the input and parameter spaces can lead to models that are both robust and generalize well, as demonstrated empirically by Wu et al. (2020), who also highlight the connection between weight flatness and the robust generalization gap. Taking another perspective on measuring smoothness, Simon-Gabriel et al. (2019) show that adversarial examples tend to correspond to regions with higher input gradients, further linking gradient norm regularization to robustness. However, this line of research typically does not address whether such regularization helps against in-manifold or off-manifold adversarial examples.

In our analysis, we connect flatness in parameter space, network Lipschitz properties, and adversarial robustness. We argue that while these defenses offer meaningful local guarantees, achieving global robustness necessitates additional assumptions about the structure and properties of the data manifold. Xuan et al. (2025) came to similar to our conclusions with respect to the effect of network confidence on adversarial robustness, but through adjusting temperature. They show empirically in preliminary experiments that high temperature in cross-entropy loss promotes better adversarial robustness. Xu & Zhang (2024) derives a measure bounding robustness generalization gap using PAC-Bayes bounds. This measure surprisingly repeats the relative flatness (Petzka et al., 2021) that we use in our derivation, confirming that such theoretical connection can be proven from different perspectives.

## F    UNNORMALIZED PLOTS

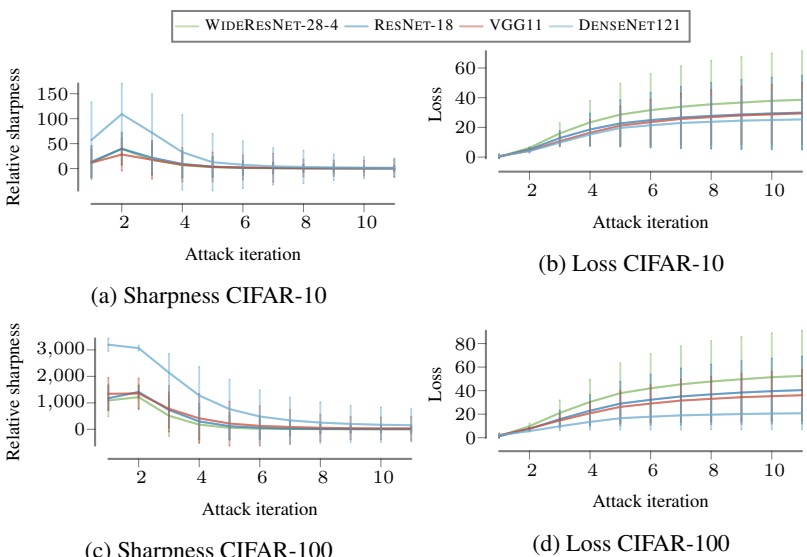

(a) Sharpness CIFAR-10

(b) Loss CIFAR-10

(c) Sharpness CIFAR-100

(d) Loss CIFAR-100

Figure 16: We report the relative sharpness on the attack trajectory of attack for WIDERESNET-28-4, RESNET-18, VGG11 and DENSENET121 on the test set of CIFAR-10 & CIFAR-100. We observe that adversarial examples first reach a sharp region, but with strength of the attack increasing they are in very flat region. We also display the standard deviation of the values on individual inputs.

## G    CODE FOR EXPERIMENTS

We use code from several resources, which we disclose here. First, the basis for training and attacking the CNNs stems from (Sehwag et al., 2020). We modified the code according to our needs. The code for DenseNet121 stems from the official PyTorch library. To attack and evaluate the LLMs, we use the official implementation of the attack (Zou et al., 2023). CIFAR-10 and CIFAR-100 were also downloaded from PyTorch.

## H    LLM USE

In this work, we used GPT-4o for both writing and coding support. On the writing side, it assisted with editing and condensing text to improve clarity. For coding, GPT-5 was used for debugging, providing autocomplete suggestions in VS Code, and generating code for LaTeX figures. Additionally, gpt-o3 was used to verify steps in the proofs and try to falsify certain steps.

