# OpenReview forum: "When Flatness Does (Not) Guarantee Adversarial Robustness"
_ICLR.cc/2026/Conference — ICLR 2026 Poster_

### Official Review · Reviewer_eTYw · 2025-10-21

**Soundness:** 3
**Presentation:** 3
**Contribution:** 3
**Rating:** 6
**Confidence:** 3

**Summary:**

This paper provides a formal analysis of the relationship between flatness in the loss landscape and adversarial robustness in neural networks. While flat minima are often believed to enhance robustness, the authors demonstrate that this connection holds only locally, not globally. They derive a closed-form measure of relative flatness in the penultimate layer and use it to constrain input-space loss variation, enabling a theoretical assessment of network robustness. Empirical results across models and datasets support these findings, showing that adversarial examples often occupy flat regions where models are confidently incorrect.

**Strengths:**

1. This work presents a framework for analyzing adversarial robustness through relative flatness, a concept introduced in prior studies.
2. While the conclusion that “flatness implies local but not global adversarial robustness” is not surprising, formally establishing this insight contributes meaningfully to the theoretical understanding of adversarial robustness.
3. The finding that adversarial examples often lie in large, flat regions is intriguing.

**Weaknesses:**

1. The theoretical analysis appears to overlook the generalization from training to unseen test data. Even if the connection between relative flatness and adversarial robustness holds on the training data, it remains unclear how flatness measured on training examples translates to robustness on unseen test inputs.
2. The paper lacks actionable insights for improving adversarial robustness, limiting its practical impact despite its theoretical contributions.

**Questions:**

1. Given the similarity between the finding that adversarial examples often lie in large flat regions and the observation of a downward trend in the input loss landscape slope (as shown by IG in Fig. 1 of [1]), it would be valuable to explore whether the theoretical framework presented in this work can explain or align with that trend. Establishing such a connection could strengthen the theoretical grounding and unify observations across studies.

2. Since adversarial training is known to significantly enhance adversarial robustness, it is important to examine how the proposed analysis interacts with or adapts to models trained adversarially. Specifically, how does relative flatness behave under adversarial training, and does the established relationship between flatness and local/global robustness still hold? Addressing this would clarify the scope and applicability of the theoretical insights.

[1] Li, Lin, and Michael Spratling. "Understanding and combating robust overfitting via input loss landscape analysis and regularization." Pattern recognition 136 (2023): 109229.

---

> ### Author Response · Authors · 2025-11-20
>
> Thank you for your thoughtful and constructive feedback and positive assessment of our formal analysis and empirical observations.
>
> **Weaknesses:**
>
> 1. Our theory is per-sample and model-conditional: for any fixed network $f$ and any point $x$ (train or test), if the penultimate-layer loss is sufficiently flat and $\phi$ is locally $L$-Lipschitz, we bound the loss increase within a radius around $x$. The proofs never assume that $x$ is a training sample, only that we measure flatness on $x$.
>
>     We do not claim that flatness measured on training data implies distribution-level robustness on the test distribution. The statistical generalization question is orthogonal: Petzka et al. 2021 have shown that under certain assumptions, flatness implies that the training loss is indicative of the test loss. Our contribution shows that flatness can guarantee adversarial robustness for the point it is measured on, but does not automatically generalize to other points. Moreover, even if flatness measured on a set of points generalized perfectly to unseen points, the best-case robustness guarantees from the Hessian would still be (i) vacuous for low-confidence points, and (ii) extremely local otherwise.
>
>     Empirically, our Uncanny Valley and scaling experiments are run on test data. That is, we train the models on a training set, but measure relative flatness on the test set and show the same phenomena (e.g., adversarial examples in large flat regions). Our findings then imply that we can derive robustness guarantees for the points we measure flatness on (both train and test), but this does not imply robustness of points we did not measure flatness on. We will clarify this scope (per-sample guarantees, including test points) more explicitly.
>
> 2. We agree that we do not propose a new defense or training strategy because; instead, our results clarify the precise limitations of Hessian-based flatness as a tool for achieving global adversarial robustness. We show that, even when the network is flat, it can be arbitrary wrong, i.e., the uncanny valley, where much of the local geometry is determined by the confidence of the network. Additionally, we show in Lemma 11 & 12, that changing the sharpness metric or loss function will not resolve this issue.
>
>     In other words, curvature as measured by the Hessian is simply too local; optimizing for flat minima in this sense can only yield mild local gains, not meaningful global robustness. We will make this perspective more explicit in the introduction and conclusion. In practical terms, our findings suggest that directly minimizing Hessian-based flatness should not be relied on as a stand-alone objective when the goal is to achieve strong global adversarial robustness.

---

> > ### Author Response · Authors · 2025-11-20
> >
> > **Questions:**
> >
> > 1. Li & Spratling (2023) analyze the input loss landscape under adversarial training and find that the input gradient norm IG (slope) decreases over training while the input Hessian spectrum HS (curvature) increases. Their IG trend is consistent with our observation that PGD trajectories end in extremely flat, high-confidence basins of the loss, the Uncanny Valley, where gradients nearly vanish. At the same time, our closed-form expression for κ links curvature to confidence, with maximal curvature near decision boundaries and flat valleys in very high-confidence regions; combined with empirical evidence that adversarially trained models are globally less over-confident than standard ones. This suggests that the rising HS in Li & Spratling can be viewed as a confidence effect: lower average confidence shifts more data toward high-curvature boundary regions even though individual adversarial examples still settle in flat Uncanny-Valley basins.
> >
> >     An interesting future direction would be to unify our feature-space flatness framework with input-space analyses such as Li & Spratling (2023). Since input gradients and curvature depend on both the classifier’s curvature and the Lipschitz/Jacobian structure of the feature extractor, a joint theory could potentially relate relative flatness in representation space to the evolution of IG and HS observed during adversarial training
> >
> >
> >     We thank the reviewer for pointing out this work and will include it in our discussion.
> >
> > 2. That is an excellent point. Adversarially trained baselines are very interesting and something we have already started to explore. In fact, we included an experiment on an adversarially trained ResNet in the appendix (“Evaluation on adversarially trained ResNet”), where we observe that adversarial training significantly increases the basin width around data points and improves local robustness. Due to space constraints, we placed this analysis in the appendix, but we will move a condensed version into the main paper and explicitly connect it to our theory. Importantly, the same local/global picture still holds: adversarial training changes the width of the basins around the training points, but it does not alter the structural limitation we prove. That is, flatness, as captured by the Hessian, can only certify robustness locally and remains insufficient to guarantee global adversarial robustness. From this perspective, adversarial training can be seen as a powerful practical mitigation that does not, however, resolve the fundamental structural limitations we identify.

---

### Official Review · Reviewer_zHuA · 2025-10-24

**Soundness:** 1
**Presentation:** 3
**Contribution:** 2
**Rating:** 2
**Confidence:** 3

**Summary:**

This paper re-assesses prior claims about the relationship between the flatness of the loss landscape and adversarial robustness. Consistent with previous claims the paper finds that flatness enhances robustness around particular training samples. However, in contrast to previous claims the current paper suggests that adversarial examples can also lie in flat regions. Hence, increasing flatness does not necessarily improve robustness.

**Strengths:**

The paper is well structured and generally clearly written.

The paper combines both theoretical and empirical research.

**Weaknesses:**

All the analysis in the paper is based on characterizing successful adversarial attacks through changes in loss rather than changes in predicted label (section 3.2). Specifically, it is claimed that an adversarial perturbation will increases the loss beyond a threshold. However, this is not true for cross-entropy loss. For example, consider a neural network that performs a 3-way classification task. If the true label of a sample is 0 and this network outputs logits [0.6, 0.1, 0.1], then the sample is classified correctly and the cross-entropy loss is 0.7944. If the sample is perturbed in such a way as to produce logits [0.6, 0.7, -5], then the predicted class label is wrong (i.e. the perturbation constitutes a successful adversarial attack), yet the loss decreases to 0.7462. The reverse is also the case: a large increase in loss does not necessarily indicate a successful attack. For example, if the same sample was perturbed so that the network produced logits [0.6, 0.5, 0.5], then the attack would be unsuccessful, but the loss would be increased to 1.0331.

**Questions:**

Given the lack of correspondence between classification accuracy and cross-entropy loss described above, which of the claims/results in the paper are still valid?

---

> ### Author Response · Authors · 2025-11-20
>
> Thank you for your thoughtful feedback. We appreciate the positive remarks on the clarity and combination of theoretical and empirical results. Below, we address our main points.
>
> You are right, the relationship between a loss increase and a prediction flip is indeed nuanced and we see how the (too) brief explanation in the text might have led to confusion. Our analysis explicitly  operates in a high-confidence regime ($\ell(x,y) < \ln 2$) and we will clearly state this in the revision.
>
> For cross-entropy loss $\ell(x,y) = -\log p_y(x)$, the condition $\ell(x,y) < \ln 2$ means the correct label has confidence $p_y(x) > 0.5$. In this case, to flip the prediction, one must reduce $p_y(x)$ to at most $0.5$, which requires increasing the loss to at least $\ln 2$. Thus, the loss must increase by at least $\ln 2 - \ell(x,y)$, and how difficult this is depends on the sharpness (curvature) of the loss surface – precisely what our derivations quantify.
>
> Outside of the high-confidence domain, when $\ell(x,y) > \ln 2$ (i.e., $p_y(x) < 0.5$), no meaningful robustness guarantee can be derived from the loss surface alone. Your counterexample illustrates this beautifully: the loss can be nearly constant, i.e., flat over a large region while the prediction flips arbitrarily, so no robustness radius can be deduced from local curvature. Rather than invalidating our bounds, your example illustrates the domain of applicability of our theory: for high-confidence samples ($\ell(x,y) < \ln 2$), relative flatness can guarantee local robustness, and for low-confidence samples ($\ell(x,y) > \ln 2$), no such guarantee can be made based solely on flatness.
>
> We will make sure to explicitly state this. In particular, we will revise the text around Definition 4, the sentence on using a “conservative threshold” to state this assumption, and to clarify that the link between loss increase and prediction change holds under high-confidence conditions and is only a sufficient (not necessary) condition for a prediction flip. Or, in short, that is, for high-confidence samples, relative flatness can guarantee robustness in a local neighborhood, and for low-confidence samples, no statement can be made.
>
> Regarding your question “Given the lack of correspondence between classification accuracy and cross-entropy loss described above, which of the claims/results in the paper are still valid?” the answer is a short **yes**. In fact, **all of our theoretical results remain valid**. In the higher-confidence regime (a regime that trained neural networks typically enter due to their very low training loss and similarly low loss on iid test data),  low sharpness is a sufficient condition for adversarial robustness, since a bounded loss implies no prediction flip. However, for the low-confidence case, it is not possible to make any statements about robustness.

---

> > ### Comment · Reviewer_zHuA · 2025-11-27
> >
> > My initial evaluation was based on a misunderstanding, and I thank the author's for their clear explanation in the rebuttal. I believe that the manuscript provides a useful analysis and makes some interesting new observations about the relationship between the loss landscape and adversarial robustness. I would therefore be happy to raise my score above the acceptance threshold, assuming the improvements that the author's have promised to myself and the other reviewers are made to the manuscript.

---

### Official Review · Reviewer_HZEM · 2025-10-29

**Soundness:** 2
**Presentation:** 1
**Contribution:** 2
**Rating:** 4
**Confidence:** 3

**Summary:**

This paper investigates the long-standing hypothesis that flat minima in the loss landscape imply increased adversarial robustness. Through a rigorous theoretical formulation, the authors show that flatness guarantees only local robustness rather than global robustness. Empirical evaluations across different architectures further support this finding, revealing that adversarial examples often reside in large, flat regions of the loss landscape.

**Strengths:**

The authors rigorously extend Petzka et al. (2021)’s notion of relative flatness to the setting of adversarial robustness, providing formal derivations that are technically sound and mathematically non-trivial. The paper presents elegant formulations that establish a clear analytical link between adversarial robustness and the Hessian of the loss.

**Weaknesses:**

1. While overall well-written, the paper is dense and somewhat derivative-heavy. Key ideas, such as the geometric mapping between feature-space and input-space curvature, could be illustrated more clearly using diagrams or intuitive explanations. A more intuitive introduction or motivating example would help readers better grasp the high-level ideas before delving into the detailed derivations. In particular, the terms “relative flatness” and “relative sharpness” are used somewhat inconsistently, which may confuse readers.

2. The empirical evaluation is somewhat limited. While the experiments illustrate the theoretical claims qualitatively, they rely on relatively weak PGD attacks (PGD-$l_2$ with $\epsilon = 0.025$ in page 8) and lack comparisons with adversarially trained baselines (e.g., TRADES, AWP, or SAM-trained models). As a result, it remains unclear whether the observed relationships between sharpness and robustness persist under stronger or more realistic adversarial conditions.

**Questions:**

1. I think the first paragraph on page 1 could be divided into multiple shorter paragraphs to improve clarity. Also, in Figure 1 (mentioned on the first page), the notation $\phi^{-1}$ is a bit confusing — it’s not immediately clear how the authors relate the feature space to the input space. It would be better to explain this mapping more intuitively in the introduction.

2. On page 2, lines 90–93, the first and second listed contributions appear somewhat redundant. In the first contribution, you state that you theoretically establish the link between flatness and adversarial robustness, while in the second, you restate this point with more detail, specifying that the link is derived through the penultimate layer. I think these two points should be combined into a single, unified contribution for clarity.

3. On page 4, in Definition 4 (Loss-change adversarial example), you define adversarial examples as those that increase the loss by more than $\epsilon$. Then, in line 183, you state that “by using a conservative $\epsilon > \log(k)$ for cross-entropy loss, we can ensure a prediction flip.” However, in practice, when the number of classes $k$ is large, this threshold can correspond to very high loss values. Such cases may represent “strong” adversarial examples (in terms of loss magnitude). Moreover, the reverse implication is not guaranteed, when $l(f(x), y)$ is not close to zero, a prediction flip might occur with much smaller loss changes. Therefore, the relationship you establish between loss increase and prediction change may not hold universally. Could you provide more discussion or clarification on this limitation?

4. On page 7, line 363, and page 8, line 378, there are two separate “Setup” paragraphs in Section 6 From Theory to Practice—one ending with a full stop and another continuing without. It would be clearer to merge them into a single, continuous setup description, as the current separation is somewhat confusing.

---

> ### Author Response · Authors · 2025-11-20
>
> Thank you for your constructive feedback. We appreciate that you value our theoretical results and recognize our derivations are sound and non-trivial. Below, we address your main comments.
>
> **Weaknesses:**
>
> 1. Thank you for acknowledging the quality of our writing. We agree that the paper is dense, in part due to the nature of our analysis, and we appreciate the reviewer’s suggestions for making it more accessible. Therefore, we agree it makes a lot of sense to explain the (geometric) mapping between feature-space and input-space curvature more intuitively in the introduction. We provided Figure 1 to give such an intuition, and appreciate your suggestions for improving it. We meant $\phi^{-1}$ as the preimage of the feature map (rather than a true inverse) and will clarify this explicitly and adjust the notation to avoid confusion. We will also revise Figure 1 and the accompanying text to better illustrate the mapping (e.g., by emphasizing the preimage interpretation and adding a simpler explanatory caption). We will also ensure consistent use of the terms “relative flatness” and “relative sharpness” by clearly defining our curvature-based measure once and using a single term throughout. (It is a bad custom in the literature to mix these up.)
>
>
> 2. We on purpose use relatively weak PGD attacks: our theory predicts that flatness only guarantees robustness locally within a basin whose radius is controlled by the curvature. Our experiments confirm this result as well as that with PGD attacks we can easily leave this robustness radius and observe the uncanny valley phenomenon. Using strong attacks would necessarily move the sample outside the local region where flatness can guarantee robustness—precisely the limitation we highlight.
>
>     That is, if we would use stronger attacks, we would lose the ability to study behavior within and at the border of the robustness radius. Phrased differently, our goal is to empirically probe the local behavior predicted by the theory, rather than to claim robustness under arbitrarily large perturbations.
>    The point about adversarially trained baselines is very interesting and something we have indeed started to explore: we include an experiment on an adversarially trained ResNet in the appendix (“Evaluation on adversarially trained ResNet”), where we observe that adversarial training significantly increases the basin width around data points. Due to space constraints, we placed this analysis in the appendix, but we will move a condensed version into the main paper and clarify the connection to our theoretical findings.

---

> > ### Author Response · Authors · 2025-11-20
> >
> > **Questions:**
> >
> > *Regarding presentation (Q1,Q2,Q4):*
> > We appreciate the stylistic suggestions on the introduction and Section 6. We will split the first paragraph on page 1 into shorter paragraphs to improve readability, and we will merge the two “Setup” paragraphs in Section 6 into a single, continuous description. We also agree that the first two contributions on page 2 are redundant; we will consolidate them into a single, unified contribution that clearly states that we establish a link between flatness and adversarial robustness via the penultimate layer.
> >
> > *Regarding Definition 4 and the link between loss increase and prediction flip (Q3):*
> > We appreciate the reviewer’s careful reading of Definition 4 and the subsequent discussion. The concern raised is valid: the relationship between a loss increase and a prediction flip is indeed more nuanced than our brief explanation in the text suggests.
> > In the current version,  the analysis operates in a high-confidence regime, and we will make this explicit. For cross-entropy loss $\ell(x,y) = -\log p_y(x)$, the condition $\ell(x,y) < \ln 2$ means the correct label has confidence $p_y(x) > 0.5$. In this case, to flip the prediction, one must reduce $p_y(x)$ to at most $0.5$, which requires increasing the loss to at least $\ln 2$. Thus, the loss must increase by at least $\ln 2 - \ell(x,y)$, and how difficult this is depends on the sharpness (curvature) of the loss surface—precisely what our derivations quantify.
> > However, when $\ell(x,y) > \ln 2$ (i.e., $p_y(x) < 0.5$), the situation is different: the prediction is already low-confidence, and it is impossible to provide a meaningful robustness guarantee from the loss surface alone. The loss can be nearly constant over a large region while the prediction flips arbitrarily, so no nontrivial robustness radius can be deduced from local curvature. This does not invalidate our bounds; rather, it clarifies their regime of applicability and fits our overarching message:
> >
> > - For high-confidence samples ($\ell(x,y) < \ln 2$), relative flatness can guarantee local robustness.
> >
> > - For low-confidence samples ($\ell(x,y) > \ln 2$), no guarantee can be made based solely on flatness.
> >
> > We will revise the text around Definition 4 and the sentence on using a “conservative threshold” to explicitly state this assumption and to clarify that the link between loss increase and prediction change holds under high-confidence conditions, and is only a sufficient (not necessary) condition for a prediction flip. That is, for high confidence samples, relative flatness can be guaranteed in a local neighborhood, and for low confidence samples, no statement can be made. Outside this regime, flatness cannot be used to give a meaningful robustness guarantee.

---

### Official Review · Reviewer_qEqm · 2025-11-01

**Soundness:** 3
**Presentation:** 3
**Contribution:** 2
**Rating:** 6
**Confidence:** 2

**Summary:**

This paper challenges the theoretical hypothesis that flatness increases the robustness of neural networks and discovered that flatness implies local but not global adversarial robustness. Flatness tends to emerge in regions where the model is highly confident.

**Strengths:**

1. The Uncanny Valley analysis helps explain why adversarial examples can appear deceptively safe, because they often lie in flat, vast, high-confidence region.
2. The paper is generally well-written and well-presented.
3. The relation between relative sharpness and adversarial robustness is clearly explained, and a precise robustness bound is given

**Weaknesses:**

1. A large part of the paper is used to justify that relative flatness at the penultimate layer is sufficient, which seems to have been stated in Petzka (2021).
2. The entire analysis was built on the local flatness at the penultimate layer; did the authors rule out the effect of the geometry of the input space on the adversarial robustness?

**Questions:**

1. Does the metric relative flatness possibly ignore the correlated curvature directions across layers?

---

> ### Author Response · Authors · 2025-11-20
>
> Thank you for the constructive feedback. We appreciate your remarks on the clarity of our presentation, the usefulness of the analysis, and the precision of our bound. Below, we address the points you raise.
>
> **Weaknesses:**
> 1. Our results indeed build on Petzka et al. (2021) but our goal and results differ significantly. Whereas they analyze how well the true distribution can be represented in a chosen layer, and informally conclude that the penultimate layer is generally a good choice, we are specifically interested in how perturbations in the input space relate to flatness in the penultimate layer. For this, we give a formal per-sample bound that links curvature in weight space to curvature in input space.
> In particular, we first derive a closed-form expression for the Hessian of the cross-entropy loss at the penultimate layer and the associated relative flatness (Eq. (3)–(4)). This makes the dependence on confidence, feature norms, and weight norms explicit. Second, we formally show (in “Why the last layer is sufficient” and Appendix C.4) that for rectified affine networks (linear/convolutional layers, BatchNorm, ReLU), all curvature originates at the loss layer and propagates backward linearly, giving a rigorous justification that the penultimate layer is sufficient for robustness analysis, not just for correlating with generalization. Third, building on this structure, we prove robustness bounds (Proposition 8 and the subsequent robustness radius bound) that explicitly relate relative flatness at the penultimate layer to a lower bound on the radius in feature and input space within which the loss cannot increase beyond a given threshold. This precise characterization of how flatness propagates through layers, and its effect on adversarial robustness is (to the best of our knowledge) completely novel.
> 2. Our analysis is intentionally local (around a specific sample) rather than global (the entire model). That is, under a flat loss surface at the penultimate layer and an $L$-Lipschitz feature extractor $\phi$, we derive a radius $\delta$ in input space such that, for any perturbation $|\xi - x|_2 \le \delta$, the increase in loss is bounded by a function of relative flatness (Proposition 8). Within this local regime, global input-space geometry does not enter directly; robustness depends on local curvature and the local Lipschitz behavior of $\phi$ around $x$.
> Our main message is that locality is a limitation: relative flatness can guarantee both generalization and adversarial robustness (which has been questioned in the literature), but 1) adversarial robustness *can only be guaranteed locally*, and 2) global robustness requires additional assumptions on the data manifold and input-space geometry. Our experiments support this, showing that adversarial examples can lie in extremely flat, high-confidence regions (Uncanny Valleys) far from actual data. We emphasize the local-versus-global distinction more explicitly.
>
> **Questions:**
>
> 1. In our setting, correlated curvature directions across layers are explicitly captured by the curvature propagation formula. For rectified affine networks, the Hessian with respect to pre-activations satisfies
>  $H_{l-1} = W_l^\top D_l H_l D_l W_l$,
>  and the weight-space Hessian block takes the Kronecker form
>  $H_{W_l} = (x_{l-1} x_{l-1}^\top) \otimes (D_l H_l D_l)$.
> Thus, curvature at earlier layers is obtained by linearly propagating the curvature at the final (loss) layer through the sequence of weight matrices and activation masks. If the final-layer Hessian vanishes, all earlier Hessian blocks vanish; conversely, high-curvature directions at earlier layers are inherited from the last layer via these linear transformations. Under our assumptions, curvature “correlations across layers” therefore do not introduce additional independent directions beyond those already present at the last layer; they are encoded in the propagated Hessian.
> The relative flatness measure at the penultimate layer thus reflects exactly those curvature directions that are most relevant for the loss, while the multi-layer geometry is summarized in the feature extractor $\phi$ and its Lipschitz constant $L$, which enter directly into our robustness bound. We explicitly focus on this rectified affine case because it is amenable to rigorous analysis as a first step.
> We agree that for architectures with non-piecewise-linear activations additional second-order contributions may arise and our analysis will need extension. We will add a brief discussion of this limitation and its implications for the learning dynamics of transformers. (In ongoing work we are exploring extensions to transformer architectures. As the attention mechanism can induce substantial curvatures when they are not saturated, analysis is rather non-trivial, however.)

---

### Author Response · Authors · 2025-12-04

Dear Area Chair,

here is the post-rebuttal status after the requested clarifications and edits have been integrated into the manuscript:

- The negative score (2) is no longer valid. That reviewer’s sole objection was the loss-prediction-flip issue. Our clarification, now explicitly incorporated into the paper, makes clear that our guarantees operate in the high-confidence regime, where a prediction flip necessarily requires a minimum loss increase. After reading this clarification, the reviewer explicitly stated "**My initial evaluation was based on a misunderstanding… I would be happy to raise my score above the acceptance threshold.**" (≥6)

- Both reviewers who gave 6 offered strong praise and had only minor clarity requests, all now addressed. Reviewers highlighted our "**precise robustness bound**" [qEqm], that our derivations are "**technically sound and mathematically non-trivial**" [eTYw], that the connection between relative sharpness and robustness is "**clearly explained**" [qEqm], and that the result "**contributes meaningfully to the theoretical understanding**" [eTYw] of adversarial robustness. Their concerns about clarity, input-space geometry, generalization framing and exposition have been resolved.

- Reviewer HZEM described our formulations as "**elegant**" and the theory as "**rigorous,**" with the paper being "**well-written.**" The reviewer raised only presentation issues, all of which have also been corrected.

- No reviewer maintains a substantive technical criticism. After revisions, all concerns, technical, conceptual, and presentation-related, have been addressed. The one reviewer who gave a reject now supports a ≥6.

Given that the revised manuscript addresses every reviewer concern and the effective post-discussion consensus is at or above weak accept.

---

### Meta-Review · Area_Chair_Z8wb · 2026-01-05

**Summary:**

The reviewers generally recognized the contribution of this work. In their initial comments, they raised substantive issues regarding presentation, experimental setup, and methodological details. The authors’ response has adequately addressed most of these concerns. For reviewer zHuA, the initial review contained some points of misunderstanding, which the authors clarified effectively in their response.

**Reviewer Concerns:**

Most of the concerns have been addressed in the rebuttal.

**Reviewer Scores:**

For reviewer zHuA, the initial review contained some points of misunderstanding, which the authors clarified effectively in their response. As a result, the reviewer may raise their score to 6.
For other reviewers, the rebuttal addressed most of the concerns, they may maintain or increase their initial scores.

---

### Decision · Program_Chairs · 2026-01-26

Accept (Poster)